# TRANSFORMERS TEND TO MEMORIZE GEOMETRICALLY; IT IS UNCLEAR WHY.

## ABSTRACT

In sequence modeling, the parametric memory of atomic facts has been predominantly abstracted as a brute-force lookup of co-occurrences between entities. We contrast this *associative* view against a *geometric* view of how memory is stored. We begin by isolating a clean and analyzable instance of Transformer reasoning that is incompatible with memory as strictly a storage of the *local* co-occurrences specified during training. Instead, the model must have somehow synthesized its own geometry of atomic facts, encoding *global* relationships between all entities, including non-co-occurring ones. This in turn has simplified a hard reasoning task involving an $\ell$-fold composition into an easy-to-learn 1-step geometric task. From this phenomenon, we extract fundamental aspects of neural embedding geometries that are hard to explain. We argue that the rise of such a geometry, despite optimizing over mere local associations, cannot be straightforwardly attributed to typical architectural or optimizational pressures. Counterintuitively, an elegant geometry is learned even when it is not more succinct than a brute-force lookup of associations. Then, by analyzing a connection to `Node2Vec`, we demonstrate how the geometry stems from a spectral bias that—in contrast to prevailing theories—indeed arises naturally despite the lack of various pressures. This analysis also points to practitioners a visible headroom to make Transformer memory more strongly geometric. We hope the geometric view of parametric memory encourages revisiting the default intuitions that guide researchers in areas like knowledge acquisition, capacity, discovery and unlearning.

## 1 INTRODUCTION

Neat representations materialize when a deep network needs to compress redundancies in data. On the other extreme, if the data is a set of atomic facts (like the birth date of a celebrity), the network would simply memorize these incompressible associations as a lookup table in its parameters (Radhakrishnan et al., 2020; Bietti et al., 2023). These two narratives have so far roughly guided our understanding of how neural networks fit sequential data. This paper fleshes out a third phenomenon in sequence modeling—glimpses of which were observed recently in Khona et al. (2024); Ye et al. (2025)—where a neat representation materializes from memorizing incompressible atomic facts. We call this *geometric* memory as it geometrically encodes *global* relationships between non-co-occurring entities. We point out that this is dramatically different from the common *associative* view of parametric memory as storing mere *local* atomic co-occurrences. From this geometry of atomic facts, we extract aspects of neural geometries that are hard to explain, posing fundamental questions about memorization in deep sequence models. To these questions, we offer some preliminary answers.

Our discussion starts at a seemingly tangential point. We begin by consolidating a fragmented set of recent demonstrations (Khona et al., 2024; Geerts et al., 2025; Ye et al., 2025; Wang et al., 2024a) that the Transformer exhibits some level of *implicit in-weights reasoning*, i.e., reasoning over knowledge from the weights without emitting an explicit chain of thought. We sharpen these results by crafting a scenario where this ability is less expected, plays out vividly, and can be cleanly isolated and analyzed. Specifically, we study path-finding on path-star graphs, a (symbolic) implicit reasoning task. The task was adversarially designed (Bachmann & Nagarajan, 2024) to cause failure of next-token trained deep sequence models—both the Transformer (Vaswani et al., 2017) and Mamba (Gu & Dao, 2023) alike. Whereas in the original task, the model is given the graph in-context, here we make the model memorize the graph's edges in its weights. Whereas in the original task, the models spectacularly fail

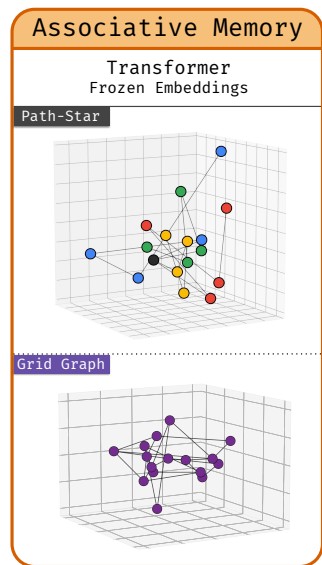
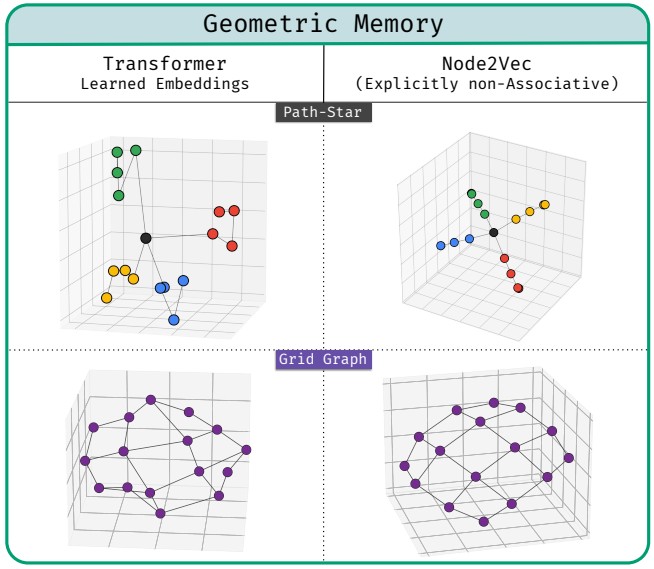

Figure 1: **Associative vs. geometric memory visualized for path-star (top) and other graphs by training a small model to memorize the edges.** Parametric memory in deep sequence models is often abstracted as if co-occurrences from atomic facts are stored in a weight matrix, while the co-occurring entities themselves are embedded arbitrarily (Wang et al., 2024b; Ghosal et al., 2024; Jiang et al., 2024a; Zhu et al., 2024; Cabannes et al., 2024b; Nichani et al., 2024; Cabannes et al., 2024a; Bietti et al., 2023; Wang & Sato, 2025). (**left**). §2.2: This *associative* view is hard to reconcile with our observation that the Transformer learns implicit reasoning on an in-weights path-star graph. §2.3: In practice, the learned embeddings (**middle**) reflect *global* structure inferred from the *local* co-occurrences, implying a *geometric* view of memory. §3: Both these views are valid ways to fit the training data, but it is not straightforward why the latter prevails during Transformer optimization. §4: When associative memory is explicitly prohibited, as in a Node2Vec model (**right**), a more elegant geometry materializes. This points to a clear headroom to improve the geometric nature of a Transformer's memory. Details of the Transformer architecture used for this visualization are provided in §D.2.2. Similar geometries for Mamba and neural networks are presented in §E.3. Also see Fig 28 for more graphs.

to learn path-finding even on small graphs, in our in-weights task, our models (both the Transformer and Mamba) succeed even on massive graphs.

The success in the in-weights path-star task, we argue, is hard to reconcile within the *associative* view of parametric memory, a convenient and highly effective abstraction of neural network memory, popular in literature (see §2.2 for references). In this abstraction (see Def. 2a), each entity is embedded arbitrarily via a "key" function $\Phi(\cdot)$, while local associations are stored in a "lookup" matrix, $\mathbf{W}_{\text{assoc}}$. The quantity $\Phi(v)^T \mathbf{W}_{\text{assoc}} \Phi(u)$ is large if *and only if* entities $u$ and $v$ co-occur during training. With such a data structure, the path-star task requires composing the aforementioned matrix operation $\ell$ times (for path length $\ell$). Unless there is step-wise supervision for each composition, learning the $\ell$-fold composition should intuitively be a daunting needle-in-the-haystack task demanding $\exp(\ell)$ time. By the specific design of our task, *every possible form of step-wise guidance is eliminated*. Yet our model appears to find the needle.

This apparent paradox begins to be resolved due to salient observations in Khona et al. (2024); Ye et al. (2025) that the node embeddings $\Phi$ reflect some notion of distance. We generalize this observation, pointing out that this has profound implications and is in fact non-trivial to explain. First, the existence of such node embeddings implies that atomic facts can be stored in an altogether distinct paradigm of parametric memory, a *geometric* one, which is in dramatic contrast to the associative one. In the simplest form of this view (see Def. 2b), the embeddings are no longer arbitrary, but rather encode the associations themselves, without relying on a separate matrix $\mathbf{W}_{\text{assoc}}$. In fact, these embeddings are so carefully arranged that they encode *global* relationships without explicit supervision to do so: even if entities $u$ and $v$ never appeared in the same context, the dot-product

$\Phi_{\text{geom}}(u)^T \Phi_{\text{geom}}(v)$ captures the model's own notion of multi-hop distance between the two. This has implications on reasoning since what seemed to be a hard-to-learn $\ell$-fold composition of local associations, now becomes an easy-to-learn 1-step geometric task for the model. We contrast the associative (Def. 2a) and geometric (Def. 2b) views of memory visually in Fig. 1 and also in Table 1.

While helping make sense of our paradox, the observed geometry raises fundamental questions. First, there must be competition between the two parametric memories, both equally valid solutions to the training objective; why does the geometric prevail over the associative? To some readers, a geometric bias may seem familiar and intuitive at first sight; but we isolate aspects that cannot be easily explained. Counterintuitively, we argue that an elegant geometry is not necessarily more succinct than a lookup of local associations. Thus, typical capacity pressures from the architecture or optimization do not explain why a highly non-trivial geometry is synthesized—that too from optimizing only over local associations. Towards understanding this, we draw connections to the simpler `Node2Vec` architecture, where we find that global geometries emerge from well-known spectral biases in such architectures. But, in contrast to prevailing theories (e.g., Levy & Goldberg (2014); see §B.6), we identify how the spectral bias arises naturally, independent of typically-assumed architectural or optimizational pressures. This gives us a preliminary insight into the natural rise of a geometric memory, albeit in a much simpler model, leaving open a foundational question for deep sequence models.

**Implications.** Although the evidence of implicit reasoning is so far limited to symbolic tasks, we believe that the geometry we isolate is a clean nucleus of geometries known to arise in language modeling e.g., (Elhage et al., 2022; Mikolov et al., 2013b; Gurnee & Tegmark, 2024). The insights from this nucleus helps conceive broader directions, practical and theoretical. First, we find that the embeddings learned by the more naive `Node2Vec` models are more strongly geometric than that of Transformers; in hindsight, this points to a well-specified headroom for making Transformer memory more geometric and less associative in practice. If the geometric bias can be improved in natural language tasks, it would also benefit natural implicit reasoning tasks where results have so far been mixed (Press et al., 2023; Biran et al., 2024; Yang et al., 2024a;b; Balesni et al., 2025). Since geometric memory encodes global relationships, it could pave the way to combinational creativity (Boden, 2003; Franceschelli & Musolesi, 2023; Nagarajan et al., 2025): discovering novel connections between information scattered in a large pretraining set. On the flip side, the interdependencies in geometric storage may impose limits on knowledge editing, unlearning and accurate retrieval. Another implication of our study is a support for why parametric memory may be superior to in-context memory, echoing Wang et al. (2024a); Geerts et al. (2025). Finally, the gap between the Transformer and `Node2Vec` geometries may also be of interest to practitioners in retrieval systems, when choosing between modern generative retrieval models (Tay et al., 2022; Wang et al., 2022; Rajput et al., 2023) and traditional dual-encoder models (Gillick et al., 2019; Huang et al., 2013). Broadly, we speculate that the associative view forms the default unstated set of intuitions that guide research in numerous areas such as knowledge acquisition, discovery, unlearning, reasoning and storage capacity; the geometric view may inspire researchers to revisit, spell out and widen these latent intuitions.

## 1.1 SUMMARY OF CONTRIBUTIONS

1. We isolate a clean, analyzable instance of implicit in-weights reasoning, and contrast the predominant local associative memory against a global geometric view of memory in deep sequence models (simple neural networks, Transformers and Mamba). (§2)

2. We show why this observation is surprising, demonstrating that the emergence of the geometric memory over the associative memory cannot be attributed to obvious architectural, optimizational or supervisory pressures. (§3)

3. We connect the global geometry to the spectral bias of `Node2Vec` dynamics and empirically intuit how it emerges without typically assumed pressures. This makes progress towards an open question in `Node2Vec` and highlights significant headroom in the embedding geometry of the current architectures. (§4)

## 2 EXPERIMENTS: IMPLICIT IN-WEIGHTS REASONING IS LEARNED

We investigate planning on a *path-star graph*, a task designed to be adversarial towards next-token learning (Bachmann & Nagarajan, 2024). The task has a clear notion of a chain-of-thought, and a

well-understood mechanism of failure for learning in-context reasoning; our hope is to repurpose this to cleanly analyze in-weights reasoning in a way that was not possible in earlier studies.

**Background.** The path-star topology (Fig. 6) consists of a root node with multiple disjoint (uniform-length) paths branching outwards. In the version of B&N'24, a model is given in context the adjacency list (with randomized node labeling and edges ordering, but fixed topology) and a goal leaf node; the task is to predict the unique path from the root to a specified goal node, without explicitly producing a chain-of-thought. To succeed, one merely needs to notice a simple right-to-left structure: the solution is the unique path back from the leaf node, reversed. Indeed, this is the implicit chain-of-thought required before emitting the first token.

Yet, on this simple task, left-to-right next-token learners are known to fail in-distribution. This failure unfolds in two stages during training: (1) On all but the first token, the model learns a trivial solution (termed a Clever Hans cheat) that is much simpler than planning: simply predict the token as the unique child of the previous ground-truth token that is revealed in the context. This crucially starves the first, key decision-making step of gradients from the rest of the path. (2) Consequently, the first token must be learned in isolation, which becomes a computationally hard learning $\ell$-hop composition problem (as detailed later). At test-time, this model defaults to guessing a random first token and continuing along that wrong path. More details of this failure are in §C.

## 2.1 In-weights path-star task.

We define an in-weights version of the above task (Fig. 7), where all examples are generated from a fixed graph $\mathcal{G}$ (of degree $d$ and path length $\ell$), rather than a fresh graph per example. Here, the model is made to memorize the full graph in its weights through edge-memorization examples, where the input is some node $v$, and the next-token target is an adjacent node $v'$. Next, path-finding examples are generated from the same graph by picking as input a random leaf node $v_{\texttt{leaf}}$ (a single token) and as target, the unique root-goal path (a sequence from $v_{\texttt{root}}$ to $v_{\texttt{leaf}}$). The model is trained for path-finding on a subset of such leaves from our fixed graph, and is tested on the remaining leaves. (We defer results on some harder variants of this topology to §E.2.)

Prior positive results of implicit in-weights reasoning are on small scales of 200 or fewer entities or on 2-hop tasks (see §B.1).Our first result below sharpens this finding: in a much larger-scale, much larger-hop path-star task that is adversarially constructed, the model learns implicit reasoning successfully.

**Setup.** We use a from-scratch, decoder-only Transformer (`GPT-mid`) (Radford et al., 2019); we corroborate all our findings on Mamba in §E.1. For the most stable results, we interleave the edge-memorization and path-finding examples during training as in Fig. 7 and also use pause tokens (Goyal et al., 2024). We found it is important to provide both forward and reverse edges for edge-memorization in order to dodge the reversal curse, but this may not be necessary for smaller graphs; see §F.1. Note that this reverse augmentation is *not* given for the path-finding examples. All details about the formatting and the hyperparameters are in §D, followed by additional analyses in §F.

**Observation 1a.** *(**Success of implicit in-weights reasoning**) On in-weights path-star graphs of as many as $5 \times 10^4$ nodes, trained on $75\%$ of the total $10^4$ paths, both the Transformer and Mamba are able to predict unseen paths when conditioned on held-out leaves with as much as 100% accuracy (see left plots of Figs. 2 and 8). Similar positive results for some harder graph topologies are in §E.2.*

Shortly, we will isolate an even stronger instance of implicit reasoning from this task for analysis. To get there, let us scrutinize where the argument of B&N'24 may go differently in the in-weights setting for the model to succeed in Observation 1a.

### 2.1.1 Where does the path-star-failure argument go wrong in-weights?

Recall that the path-star failure unfolds in two stages. Perhaps one of these stages does not play out in the in-weights setting. A first possibility could be that the Clever Hans cheat is not picked up here. The cheat was a simple, left-to-right pattern that fits all but the first token as the unique neighbor of the preceding ground-truth token that was present as input. Such left-to-right cheats are simpler than the true right-to-left solution, and thus quickly learned. However, perhaps when the target we train on is an *in-weights* path, the cheat is not easy to learn—say, due to the nature of recalling from parametric memory. A complex cheat may not be learned quickly, allowing gradients from future

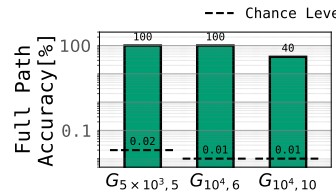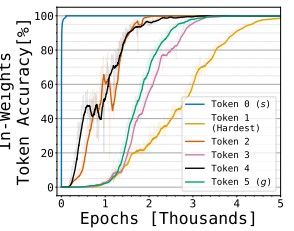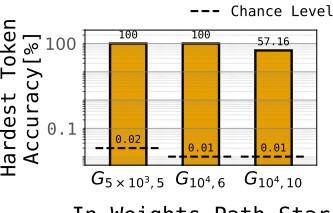

Figure 2: **Success of Transformer in in-weights path-star task. (left)** A next-token-trained Transformer achieves perfect or highly non-trivial accuracy on large path-star graphs $\mathcal{G}_{d,\ell}$ (Observation 1a). **(middle) Learning order of tokens.** The tokens of a path are not learned in the reverse order i.e., the model does not learn the right-to-left solution. Thus, gradients from the future tokens are not critical for success (Observation 1b). **(right) Success of hardest-token-only task.** In fact, the hardest token (the first) given the leaf is learned in isolation to non-trivial accuracy (Observation 1c). Success of this $\ell$-fold composition task is hard to explain within the associative memory view (§2.2). Analogous plots for Mamba are in Fig. 8. We invite the reader to contrast these in-weights task results with the in-context task ones in Fig. 5.

tokens to reach the first token representation, in turn allowing the correct, right-to-left solution to compete and emerge. We summarize this hypothesis out below:

**Hypothesis 1a.** *(Model may experience future-token gradients) The in-weights path-star task is solved (Observation 1a) because Clever Hans cheats are not learned quickly enough, allowing future-token gradients to persist, teaching the model to find the right-to-left solution.*

We can indirectly test for this hypothesis as follows. If the model learned the right-to-left dependencies, it must also learn the tokens in the reverse order: the unique predecessor of the goal node is the easiest to identify (under the right-to-left dependencies) and will thus be learned first; the next-easiest is the next predecessor and so on, until the first node (which depends on all that has been learned so far). Indeed, in the in-context setting of B&N'24 where they explicitly switch off the Clever Hans cheat (under teacherless training), such a reverse-learning cascade is observed (Fig. 5). However, we do not see this in the in-weights setting.

**Observation 1b.** *(No reverse-learning cascade) The target tokens in the in-weights path-star task are learned in no particular order by the Transformer and Mamba models (see middle plots of Figs. 2 and 8 in contrast with Fig. 5; or see Fig. 24 for side-by-side comparison).*

The above observation weakens Hypothesis 1a, encouraging us to search for another one. A second possibility is that the second stage of the failure of B&N'24 does not trouble us in the in-weights task. Recall that in this stage, the first token is to be learned in isolation without future-token gradients. But this is an $\ell$-fold composition task, that is theoretically well-understood to be computationally hard (as we elaborate later). Perhaps, that is not the case for our in-weights task:

**Hypothesis 1b.** *(First-token may be easy to learn) Learning the key decision-making token (the first token) in the in-weights path-star task is not computationally hard.*

Testing Hypothesis 1b is easy: we train the model simply on the first token loss, instead of the loss over the full path sequence. We find that this task is trivial here, affirming Hypothesis 1b, and isolating a much stronger and cleaner instance of implicit in-weights reasoning, one that shortly leads us to our main insight.

**Observation 1c.** *(Hardest, first token is learned in isolation) In the in-weights path-star task with edge-memorization and only first-token-training examples, both the Transformer and Mamba models learn the first token in isolation—without any intermediate supervision from other nodes in the path (Figs. 2 and 8 (right)).*

## 2.2 THE CONTRADICTION BEHIND LEARNING THE HARDEST TOKEN

The fact that the hardest (first) token is learned (as in Observation 1c), we argue, is difficult to square with the abstraction that parametric memory strictly stores local associations. Concretely, in this abstraction, the first node requires composing a local associative recall function $\ell$-many times: recall the predecessor of $v_{\texttt{leaf}}$ from the weights, namely $v_{\ell-1}$, then the predecessor of that, $v_{\ell-2}$, and so on. In short, one can write $v_1 = \texttt{Predec}^\ell(v_{\texttt{leaf}})$. This recall function takes the shape of a matrix operation as described below:

**Definition 2a.** *(Local associative parametric memory) We say that a deep sequence model $f$ has memorized a graph $\mathcal{G} = (V, E)$ associatively iff for any vertices $u$ and $v$, $f(u)[v]$ is high only on adjacent vertices i.e., for $(u, v) \in E$. This model is typically abstracted as $f(u)[v] = \mathbf{\Phi}(w)\mathbf{W}_{\texttt{assoc}}\mathbf{\Phi}(v)$; here, $\mathbf{W}_{\texttt{assoc}}$ encodes the associations as $\sum_{(u,v) \in E} \mathbf{\Phi}(u) \otimes \mathbf{\Phi}(v)$; the embeddings $\mathbf{\Phi}$ are arbitrary "keys" that by themselves encode no associations in graph $\mathcal{G}$ e.g., they are modeled as orthogonal or random embeddings in literature.*

With this structure however, the first-token $\ell$-fold composition task should intuitively require $\Omega(\exp(\ell))$ compute to learn—just like the first token was demonstrably hard-to-learn in the in-context path-star task (B&N'24). This hardness can be theoretically formulated in many ways (see §B.2); one way to intuit this is to notice that the optimization task is a search for a needle in the haystack: there is an $\exp(\ell)$ space of possible discrete compositions, and all but the correct one have an equally miserable loss value; with the loss terrain rendered flat and the gradients uninformative, the learner is forced to sift through the vast hypothesis space to find a needle. As a preliminary corroboration, in §G.1, we find that models fail at this task when the embeddings are frozen.

This barrier could be surmounted if, rather than providing supervision from only the end output of the composition, there was supervision from each hop, which would cast a graceful loss landscape. This could mean providing a data curriculum or chain-of-thought supervision or reducing compositionality through various means.Indeed, prior positive results on implicit in-weights reasoning involve one such aid or the other.(see §B.1). Our setting, by design, offers none of these aids. Our paths have fixed lengths (hence, no implicit curriculum), are disjoint (so, no test-train overlaps), and require as much as a 10-fold composition (so, a daunting exponent). What we isolate, therefore, is a stronger and more sterile instance of implicit in-weights reasoning, one that is cleanly inconsistent within the associative view of parametric memory.

## 2.3 TWO COMPETING DATA STRUCTURES FOR PARAMETRIC MEMORY

The paradox begins to resolve with observations in Khona et al. (2024); Ye et al. (2025) that we will flesh out in our setting. Both find that the nodes of their (smaller) graphs are embedded in a way that reflects a notion of distance, aiding their respective implicit reasoning tasks. Before extending this to our much larger graphs, we point out that this observation has profound implications. First, this presents an altogether different view of parametric memory itself: a memory of atomic facts that does not take shape as an associative lookup (over arbitrary embeddings) but as a geometry of highly-organized embeddings. Next, importantly, whereas an associative memory only makes local information accessible, a geometric memory readily betrays global multi-hop relationships—even when trained only on local associations, as we establish shortly. We define this below, and compare these two forms of memory in Table 1.

**Definition 2b.** *(Global geometric parametric memory) We say that a deep sequence model $f$ has memorized a graph $\mathcal{G} = (V, E)$ geometrically iff for any pairs of vertices $u$ and $v$, $f(u)[v]$ reflects some notion of multi-hop closeness in $\mathcal{G}$. This can be abstracted as $f(u)[v] = \mathbf{\Phi}_{\texttt{geom}}(w) \cdot \mathbf{\Phi}_{\texttt{geom}}(v)$, where $\mathbf{\Phi}_{\texttt{geom}}$ reflects some graph embedding of the vertices.*

**The resolution.** One can view the two parametric memories of Defs. 2a and 2b as two competing data structures, both representable by a deep sequence model, each yielding its own learning complexity of the hardest token (much like how a heap or an array would yield different search complexities). The local associative memory is analogous to a linked list: for the first token, this would incur a (benign) $\ell$-hop lookahead step but *learning* this incurs an exponential cost. The geometric memory is powerful in that it reduces this learning complexity all the way to $\Theta(1)$. For instance, we may expect that the embeddings of all nodes in path $i$ are clustered tightly around a unique path vector

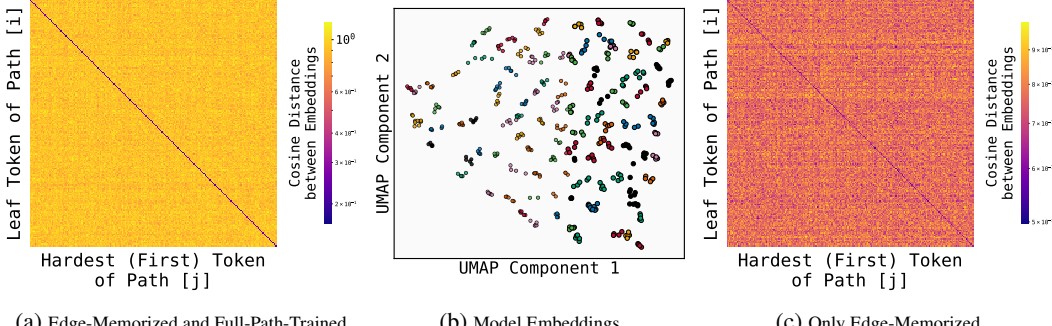

(a) Edge-Memorized and Full-Path-Trained      (b) Model Embeddings      (c) Only Edge-Memorized

Figure 3: **Evidence of global geometry of Transformer in path-star task.** **(a)** In the heatmap, entry $(i, j)$ is the cosine distance between the *leaf* embedding of path $i$ (row) and the *first-hop* embedding of path $j$ (col). The clear diagonal line implies that embeddings within each path are more aligned, reflecting global structure. **(b)** `UMAP` projection of token embeddings where each point is a node embedding; color indicates path identity. Different paths form separated clusters (see Fig. 18 for clearer image). **(c)** For a later experiment in §3.2: heatmap for a model trained only on edge-memorization still reveals a level of geometry, although weaker than in (a). Analogous plots for Mamba are in Fig. 9

$\mathbf{z}_i$. In this data structure, learning the first token given the leaf node is a one-hop task: simply find a one-hop neighbor of the central node that is best-aligned with $\mathbf{\Phi}_{\mathtt{geom}}(v_{\mathtt{leaf}}^{(i)})$. Such a structure indeed materializes in our massive graphs:

**Observation 2.** *(Evidence of global geometry) For our large path-star graphs, the (token) embeddings of the leaf and first node of a path are clustered closer to each other than those of other paths in both the Transformer and Mamba (Fig. 3 and Fig. 9). We corroborate similar geometries for a variety of small graphs in Fig. 1 and Appendix E.3 through direct visualizations.*

## 3 THE EMERGENCE OF GEOMETRIC MEMORY IS NOT EASY TO EXPLAIN

In this section, we demonstrate that the geometry cannot be easily explained by well-understood learning pressures, be it from the architecture, the optimizer, or the supervision. During training, the two types of parametric memory—two equally valid ways to fit the training data—must compete with each other. Depending on the reader's prior, the rise of the geometric memory may seem familiar especially in light of well-known geometries of high-level concepts and features. While this impression is valid in part, we will carefully isolate aspects that are unexpected.

### 3.1 CAPACITY PRESSURES DO NOT EXPLAIN GEOMETRIC MEMORY.

In theories of generalization (Neyshabur et al., 2015; Zhang et al., 2017), representations arise from pressures to represent the data as succinctly as possible, precluding a brute-force lookup. These capacity pressures are either explicit (e.g., the architecture or the regularizers) or implicit (e.g., biases of the gradient descent optimizer). For instance, a word may co-occur with thousands of similar contexts in training. A lookup of those specific contexts is too cumbersome. But fortunately, much redundancy exists in the data. Such redundancies are compressed away by the learner. From this arises a succinct, elegant geometry of high-level embeddings—such as the ones witnessed in neural word embedding models (Mikolov et al., 2013a). Perhaps, a parallel logic explains our observations.

**Hypothesis 2a.** *(Explicit or implicit capacity pressure) Associative memory is either explicitly impossible to represent—due to the parameter count or the design of the architecture, especially the embedding bottleneck—or is implicitly less preferred by gradient descent. This is because geometric storage is a more succinct representation of the data than associative storage.*

This logic can be dismantled as follows. First, we show that the learner *can* learn the associative model, thus questioning the role of explicit pressures. Positive theoretical results in Nichani et al. (2024) (see their Theorem 2) roughly prove that, with $m$ (frozen) embedding dimensions, the $m^2$

parameters in $\mathbf{W}_{\text{assoc}}$ can store $m^2$ many associations. Our models have $m \approx 400$ which appears reasonable to represent our large graphs associatively. However, this expressivity result is not a proper refutation: the results of Nichani et al. (2024) cannot be rigorously imported to us, and besides, there may be yet other explicit pressures that discourage associative memory during optimization (e.g., normalizations, weight decay). As a definitive refutation, we make an empirical demonstration: there are optimization settings where geometric memory naturally arises, but in those very settings, an associative memory can be learned with the embeddings frozen and all else unchanged. Thus, although an associative memory is artificially possible, a geometric one had naturally emerged.

**Observation 3a.** *(Associative memory can be artificially learned with our architectures) On various tiny graphs and various sequence models (Transformers, Mamba, neural networks) Figs. 1 and 28, a geometric memory arises naturally even though the same setup learns to set the data with frozen embeddings. This storage is purely associative since only one trainable layer exists (see §D.2.2).*

Thus explicit capacity pressures do not force a geometric memory in place. But perhaps, a geometry still arises due to implicit pressures from gradient descent to represent succinctly. To refute this, we refute the underlying premise itself: contrary to intuition, a geometric storage is not necessarily more succinct than an associative storage. To see why, we first clarify that unlike a typical natural language task, our setting must be recognized as a *memorization* task (in the sense of Feldman (2020)): the existence of an edge cannot be statistically surmised from the rest of the training set. In other words, while typical word embedding tasks contain statistical redundancies—which when compressed give way to simple patterns—no such redundancies exist in our task. Thus, we will show, for certain graphs, straightforward notions of succinctness does not break the tie between the two ways of representing the data. Concretely, in generalization theories, the (bit or norm) complexity of a lookup table is larger than the succinct one *by a factor that scales polynomially with the size of the training set.* In contrast, in the path-star or cycle memorization task, *this gap reduces to a constant* (that can lie in $[1, 2]$ *independent of graph size*), implying a weak to no implicit capacity pressure:

**Proposition 1.** *(Geometric and associative memory are roughly equally succinct) For certain graphs, the complexity of the local associative memory both in terms of bits and $\ell_2$ norms is either equal to or at most twice that of the global geometric memory (where equality is achieved if there is no weight-tying or if only forward edges are stored. (Proof and more explanations in §G.2)*

### 3.2 SUPERVISORY PRESSURE DOES NOT EXPLAIN GEOMETRIC MEMORY

Orthogonally, one could attempt to attribute the global geometry to the "global" supervision itself:

**Hypothesis 2b.** *(Global supervision may explain global memory) A global geometry arises over local associations since the training involves a (global) path-finding objective.*

This hypothesis however is already weakened by design in the path-star task. First, path-finding supervision is not provided for unseen paths, paths which nevertheless exhibit a global geometry. Next, in the hardest-token-only task, path-finding supervision is applied only on the path end-points; yet a geometry materializes on intermediate nodes (see Fig. 16b). We give an even cleaner refutation of this hypothesis by analyzing models trained purely on local supervision (i.e., on edge memorization):

**Observation 3b.** *A global geometry emerges even in locally-supervised models as seen in the embeddings of tiny graphs of various architectures in Fig. 1 and §E.3 and in the heatmaps for the large path-star graph (Figs. 3c and 9c). A locally-supervised model can be subsequently finetuned purely on the hardest-token task and achieve high test accuracy on path-finding (§F.3).*

## 4 GEOMETRY ARISES FROM NATURALLY-OCCURRING SPECTRAL BIAS, WITHOUT PRESSURES

Setting aside the competition between the two parametric memories, we can still extract a non-trivial question: how does gradient descent synthesize global information from mere local supervision, without various pressures, and what geometry does it produce? To isolate this, we turn to simpler, 1-layer, 1-hop `Node2Vec` models. These are equivalent to a Transformer, trained only on

edge-memorization, with only an embedding and unembedding layer—thus, associative memory is architecturally prohibited.

A precise characterization of what embeddings are learned even in such simple models is an open question, but a rich line of work (albeit with key assumptions about various pressures outlined shortly) points to a *spectral bias*: the learned embeddings often align with the top (non-degenerate) eigenvectors of the negative graph Laplacian. This indeed holds: the `Node2Vec` embeddings in Fig. 1 right column matches the top eigenvectors—called the Fiedler vector(s)—in Fig. 4 left column. This leads us to a few key insights. First, these eigenvectors happen to be the very source of global geometries. But secondly, these `Node2Vec` geometries turn out to be more well-organized than what the Transformer exhibited (in Fig. 1 middle column). Thus, we conjecture a similar but—in hindsight—somewhat "adulterated" spectral geometry in Transformers:

**Hypothesis 3.** *(Spectral bias) A Transformer memorizes with a global geometry due to spectral biases; but there is significant headroom in the quality of geometry, likely because the representation is adulterated with local associative memory.*

A natural next question is to wonder where the spectral bias stems from. Going back to `Node2Vec` theories (see §B.6), the literature suggests that similar models rely on the top eigenvectors due to the explicit pressures from a bottleneck or regularization or a multi-hop supervision. Our setting defies all these assumptions.

A further discrepancy in existing analyses is that they are on simpler, non-cross-entropy losses. In these systems, the dynamics simplify nicely yielding a closed-form solution for the inner products of the embeddings. However, we use the cross-entropy loss (to be faithful to how sequence models are trained), under which an expression for the inner product is evasive. The dynamics here may behave in one of many complex ways: it may simply diverge, or it may converge in direction (like in logistic regression (Soudry et al., 2018)); the converged direction in turn, may be degenerate or not. Our empirical analysis points to a special dynamic: the system does converge to a meaningful zero-gradient solution, working its way towards a neat two-fold property, while exhibiting a spectral bias naturally:

**Observation 4.** *(How spectral bias emerges without typical pressures) In a $1$-layer, $1$-hop* `Node2Vec` *model with the embedding $\mathbf{V} \in \mathbb{R}^{n \times m}$ of $n$ nodes, with the dynamics denoted as $\dot{\mathbf{V}}(t) = \eta \mathbf{C}(t)\mathbf{V}(t)$ (where $\mathbf{C}(t) \in \mathbb{R}^{n \times n}$ is a time-dependent co-efficient matrix), the converged solution for our small graphs is such that (a) the columns of embedding matrix $\mathbf{V}$ span the graph's Fiedler-like vectors, and (b) the co-efficient matrix $\mathbf{C}$ has those same vectors in its null space (Fig. 4). Crucially, this dynamic does not need a low rank constraint; the embedding size $m$ can be larger than the graph.*

We provide in §H an empirically-informed intuition for a "self-stabilizing" dynamic that gradually filters out lower eigenvectors; we leave open a formal analysis. Admittedly, neither is this analysis on a deep sequence model, nor does it divulge anything about the competition between associative and geometric memories. What it does is make progress on an open question for a simpler model. It also gives us an idea of how and what global information can arise naturally out of local supervision, devoid of any supervisory, bottleneck-driven, or regularizing pressure.

## 5 RELATED WORK

We briefly discuss key related lines of work here, deferring details and many other lines to §B.

**Analysis of Transformer memory** Various works have analyzed how neural networks implicitly behave as associative memory stores, such as in autoencoders (Radhakrishnan et al., 2020) or Transformers (Bietti et al., 2023; Cabannes et al., 2024a; Nichani et al., 2024; Geva et al., 2021; Schlag et al., 2021; Sukhbaatar et al., 2019). We emphasize that the associative memory view is sufficient to understand Transformer behavior on disjoint facts, evidenced by the rich empirical literature built on this view. There have also been mechanistic studies how Transformers perform fact recall (Lv et al., 2024; Geva et al., 2023; Meng et al., 2022) and where memory is stored (Geva et al., 2021). Directly related to us are the works of Khona et al. (2024); Yao et al. (2025); Biran et al. (2024) who perform an interpret multi-hop recall circuits.

**Multi-hop question-answering** A line of work has looked at natural language based two-hop questions on pretrained models e.g., "What is the calling code of the birthplace of Frida Kahlo?" (example

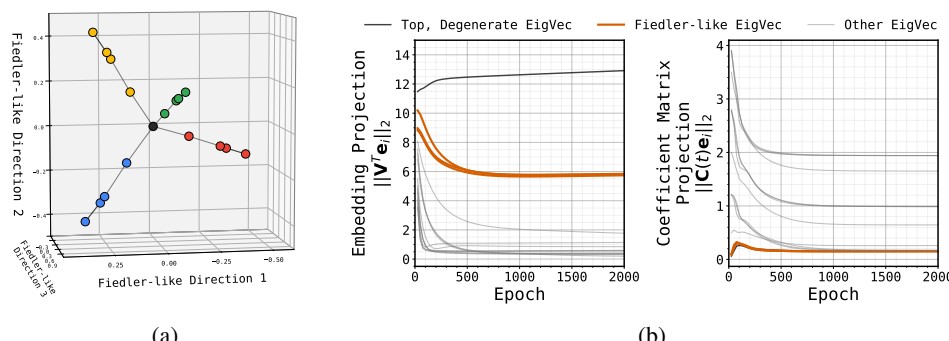

(a)  (b)

Figure 4: **Spectral geometry arises in `Node2Vec` even without low-rank pressure (Observation 4) for Path-Star Graph**. (a) The graph's Fiedler vectors shown here closely mirrors the UMAP directions of the `Node2Vec` embeddings in Fig. 1 (right). (b) The evolution of eigenvector projections during training. *(left)* The embedding matrix $\mathbf{V}$ aligns with the Fiedler-like eigenvectors (and a top, degenerate vector) evidenced by the projection norm $||\mathbf{V}^T\mathbf{e}_i||_2$ converging to a stable, non-zero value. Projections of other eigenvectors diminish towards zero. *(right)* Concurrently, the Fiedler-like eigenvectors move into the null space of the co-efficient matrix $\mathbf{C}$ in that the norm $||\mathbf{C}\mathbf{e}_i||_2$ converges to 0. Crucially, this spectral bias arises without a low dimensional constraint ($d_{\text{embedding}} = 100$, much larger than nodes in graph).

from Press et al. (2023)). Results here have been limited (Press et al., 2023; Biran et al., 2024) or mixed (Yang et al., 2024a;b; Balesni et al., 2025). Yao et al. (2025); Wang et al. (2024a) find that these queries require exponential amounts of data, or a curriculum, or very long amounts of training. A dedicated study of the gap between these and the synthetic settings is important and left for future work.

**Analysis of embedding geometries.** Exactly what embeddings are learned by various contrastive losses such as in `Word2Vec` and `Node2Vec` has been a well-studied open question. Most other analyses are on a simpler variant called the negative sampling loss where the closed form expression for the inner products is relatively straightforward (Levy & Goldberg, 2014). But this expression does not tell us what the embeddings themselves are. The connection to the graph spectrum has been established in settings with explicit multi-hop objectives like DeepWalk (Qiu et al., 2018), in low-rank settings (HaoChen et al., 2021; Tan et al., 2024; Jaffe et al., 2020) or quadratic losses with early stopping (Karkada et al., 2025). Other analyses of `Node2Vec` focus on specific types of graphs like stochastic block models (Davison et al., 2024; Harker & Bhaskara, 2024).

## 6 CONCLUSIONS AND FUTURE WORK

While the associative view of parametric memory is a simple and highly effective view of neural networks, we demonstrate that a geometric view of storage is necessary to explain the behaviors of deep sequence models. Various practical questions arise out of these findings. First, a geometry makes learning implicit reasoning possible. However, implicit reasoning in natural language have so far been mixed (Press et al., 2023; Biran et al., 2024; Yang et al., 2024a;b; Balesni et al., 2025), suggesting that we may need to improve geometry in natural language tasks. Indeed, the elegant geometries of `Node2Vec` models indicate that Transformer memory can be made much more geometric. Orthogonally, our findings are of relevance to making choices between parametric vs. contextual memory, and also between generative retrieval vs. dual encoder retrieval models. Our work raises the foundational question of when and how associative and geometric memory compete with each other during optimization, and what factors—such as graph connectivity, training time, learning rate—can foster one over the other. Broadly, we expect our findings to inspire revisiting unstated associative assumptions underlying research on knowledge and memory in language models. We discussion limitations in §A.

**Ethics statement.**  This work is purely conceptual and theoretical in nature and the authors acknowledge the ICLR code of ethics. No human subjects, personal data, sensitive information are involved.

**Reproducibility statement.**  We have taken several steps to ensure reproducibility of our results. (1) *Experimental design:* The in-weights path-star task and its in-context counterpart are formally described in Section 2, with additional details on training regimes and failure cases of the in-context task in Appendix C. (2) *Model setup:* Architecture choices (`GPT-mid`), training schedules, hyperparameter settings, tokenization, prefix/target formats, and the role of pause tokens are provided in Appendices D and F. (3) *Analysis:* Sensitivity analyses and extended empirical results, such as the hardest–token–only setting and embedding geometry analyses, are presented in Appendix F. Informal propositions (e.g., spectral bias) are labeled as propositions Section 4, with informal proofs and extended discussion in Appendices G.2 and H.

**LLM usage statement.**  We declare the use of Large Language Models (LLMs) as general-purpose assistive tools in the preparation of this manuscript. LLMs were used for minor code refactoring and visualization adjustments, and helping locating some relevant literature () with all sources subsequently manually verified. No part of the conceptualization, theoretical contributions, experimental design, or core scientific results relied on LLMs. The originality, technical content, and analysis presented in this work are entirely the result of the authors' efforts.

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

Table 1: Comparison of two forms of parameteric memory in a deep sequence model, given a dataset of edges from a graph $\mathcal{G} = (V, E)$

| | **Associative Memory** (Def. 2a) | **Geometric Memory** (Def. 2b) |
|---|---|---|
| **Storage mechanism** | A lookup table of pairwise edge associations | A graph embedding of the vertices |
| **Embedding $\boldsymbol{\Phi}$** | Arbitrary (random or orthogonal); acts as keys for lookup; doesn't reveal the associations in itself. | Reveals the graph structure. |
| **Simplest neural implementation (of the learned similarity between nodes $u$ and next-token $v$, denoted by $f(u)[v]$)** | $f(u)[v] = \boldsymbol{\Phi}(u)^T \mathbf{W}_{\mathtt{assoc}} \boldsymbol{\Phi}(v)$ where $$\mathbf{W}_{\mathtt{assoc}} \approx \sum_{(u,v) \in \mathcal{G}} \boldsymbol{\Phi}(u) \otimes \boldsymbol{\Phi}(v)$$ and $\boldsymbol{\Phi}(u) \cdot \boldsymbol{\Phi}(v) \approx 0$ for $u \neq v$. | $f(u)[v] = \boldsymbol{\Phi}_{\mathtt{geom}}(u)^T \boldsymbol{\Phi}_{\mathtt{geom}}(v)$ where $\boldsymbol{\Phi}_{\mathtt{geom}}$ corresponds to graph Laplacian. |
| **Global information** | Only local edge information is readily available i.e., $f(u)[v]$ high only for $(u, v) \in E$ | Readily available i.e., $f(u)[v]$ is proportional to multi-hop distance between $u$ and $v$ in $\mathcal{G}$ |
| **Novel information not present in data** | None; only stores seen associations. | Has synthesized novel information by integrating information scattered across multiple datapoints. |
| **Implicit compositional reasoning e.g., path-finding** | Intuitively, exponentially hard to learn. | Approximated as an easy-to-learn geometric navigation task. |
| **Accuracy of retrieval** | Precise | Approximate |
| **Description length** | Number of edges, $|E|$ | Number of vertices times the embedding dimensionality, $|V|m$ |

# APPENDIX

## A LIMITATIONS

1. Our positive result of implicit in-weights reasoning is on a purely symbolic task, and on a specific graph topology (path-star, and tree-star in §E.2). It is unclear how well this generalizes to other topologies, and to graphs of other sizes.

2. Whether our insights extend to natural language is highly non-trivial, since the way the entities are tokenized and the way relationships are presented are much more unstructured.

3. We use small to mid-sized Transformers trained from scratch (GPT-mid). We have not explored the effect of large model sizes or of large-scale pretraining.

4. We emphasize that all our arguments (e.g., about the lack of pressure) are empirical and informal. Perhaps, a slightly more nuanced form of architectural or statistical pressure (e.g., more nuanced norm complexity such as flatness of the loss) may indeed explain why associative memory is less preferred by the model. Conversely, perhaps the lack of pressures in the learning setup is indeed why the Transformer learns a sub-optimal kind of geometric memory compared to Node2Vec.

5. Although we illustrate a clear contrast between associative and geometric memory in their caricatured forms (i.e., as $\boldsymbol{\Phi}(u)^T \mathbf{W}_{\mathtt{assoc}} \boldsymbol{\Phi}(v)$ vs. $\boldsymbol{\Phi}_{\mathtt{geom}}(u) \cdot \boldsymbol{\Phi}_{\mathtt{geom}}(v)$), it is unclear how to conceptually disentangle these two modes of storage in a given multi-layered deep network.

## B  RELATED WORK

Our work consolidates fragments of a nascent phenomenon and weaves together distinct lines of theoretical and empirical work on memorization, learning compositional functions, and interpreting model representations. We elaborate on each of these threads below.

### B.1  IN-WEIGHTS REASONING TASKS

**Synthetic graph tasks.** Our work consolidates positive results of in-weights reasoning in literature, and presents a stronger instance of it, removing various confounders that make composition-learning easy. Khona et al. (2024) report successful path-finding on 200 nodes-large in-weights graphs, with varying path lengths and test-train overlap. Ye et al. (2025); Wang et al. (2024a) report positive results on much shorter 2-hop tasks over 1000 entities. Geerts et al. (2025) look at in-weights transitive inference, a special type of $\ell$-fold composition query where the model is trained on local comparisons and is queried on more distant comparisons. On settings with 7 objects, Geerts et al. (2025) find a clear difference between an in-context version of this task (where the model struggles) and an in-weights version (where the model succeeds). Such relational queries are equivalent to giving two nodes along a path and querying which node is closer to the center of the graph. Our task of finding the first node is a much harder *search* task, where one finds the smallest node in an ordered relationship. Nagarajan et al. (2025) discuss the limitations of next-token prediction, including on open-ended in-weights tasks, in lower data regimes. The fact that their next-token predictor achieves non-trivial performance on their in-weights task could be attributed to the effects of a geometric memory. Tangentially related is the positive finding in Yin & Wang (2025) that the Transformer can compose in-weights knowledge given in-context demonstrations. It is worth noting that the (theoretical) arguments in both these works (Nagarajan et al., 2025; Yin & Wang, 2025) rest on the associative memory view.

As a negative result, Wang et al. (2024b) report that, on in-weights graphs of less than 500 nodes, models are only able to infer already-seen paths or sub-paths, but not beyond them. We suspect this may stem from the fact that their model is only trained on the paths themselves (while our work and Khona et al. (2024) make the model memorize edge bigrams).

**Multi-hop question-answering.** A line of work has looked at natural language based two-hop questions on pretrained models e.g., "What is the calling code of the birthplace of Frida Kahlo?" (example from Press et al. (2023)). Results here have been limited (Press et al., 2023; Biran et al., 2024) or mixed (Yang et al., 2024a;b; Balesni et al., 2025). Yao et al. (2025); Wang et al. (2024a) study multi-hop queries on synthetic knowledge (with the former on 2-hop queries while the latter explore upto 4 hops) and find that these queries can be learned provided there is exponential amounts of data, or a curriculum, or very long amounts of training. Balesni et al. (2025) report that models are unable to compose synthetic facts, but can succeed in composing a synthetic fact with a natural one. Orthogonally, Wang et al. (2025a) identify that such in-weights implicit reasoning can be hurt by scaling up the parameters. Perhaps some of these negative results may be attributed to the reversal curse (Berglund et al., 2024; Allen-Zhu & Li, 2023), or the lack of extended computation e.g., we use pause tokens (Goyal et al., 2024). A dedicated study of this gap between our synthetic settings and these settings is important and left for future work.

**Reversal curse and (a)symmetric knowledge.** The reversal curse (Berglund et al., 2024; Allen-Zhu & Li, 2023) is a well-known out-of-distribution, in-weights failure mode of next-token-trained Transformers. Such models are unable to recall $u$ given $v$, when trained to recall $v$ given $u$, suggesting asymmetric storage in parametric memory. Fixes for the reversal curse have involved reversed or permuted data augmentation (Guo et al., 2024; Lu et al., 2024; Kitouni et al., 2024; Golovneva et al., 2024). In our settings, we find that reverse edges are critical to elicit implicit reasoning in our path-star tasks (see §F.1), but is not necessary to elicit a geometry in the smaller graphs.[1] Various theories have been proposed to understand this failure (Lin et al., 2024; Zhu et al., 2024; Wang & Sun, 2025), which may be worth revisiting under a geometric view. One may also view our contrast between associative and geometric memory (of a generic graph data) as a generalization of the aforementioned contrast between the asymmetric and symmetric knowledge storage (of a more specific, disjoint set

---

[1]It appears that the observations on small graphs in Khona et al. (2024) do not require memorizing the reverse edges, which aligns with our findings on small graphs.

of associations). We leave it for future work to discover a nuanced connection between our work and the reversal curse.

## B.2 Failure of end-to-end composition learning

While $\ell$-fold composition functions are surprisingly easy to *express* in transformers (Sanford et al., 2024b), empirical results have time and again demonstrated that they are hard to *learn* through gradient-based methods, both in traditional deep network settings (Shalev-Shwartz & Shashua, 2016; Abbe & Sandon, 2020; Gülçehre & Bengio, 2016; Glasmachers, 2017; Abbe & Boix-Adsera, 2022; Abbe & Sandon, 2020) and more recently in language models too (Nye et al., 2021; Ling et al., 2017; Cobbe et al., 2021; Piekos et al., 2021; Zelikman et al., 2022; Recchia, 2021; Cobbe et al., 2021; Hsieh et al., 2023; Shridhar et al., 2022). Others (Bachmann & Nagarajan, 2024; Hu et al., 2025; Shalev-Shwartz & Shashua, 2025) demonstrate how next-token learning can trap training at a stage where composition learning becomes a problem. Theoretical works have attempted to formalize these failures by demonstrating limits due to to expressivity (Malach, 2023; Chen et al., 2024; Peng et al., 2024) or sample complexity (Shalev-Shwartz & Shashua, 2016) or computational complexity (Wies et al., 2023; Hu et al., 2025) or in terms of statistical queries (Wang et al., 2025b). These results do not prove that a fixed composition function cannot be learned—only that a worst-case function exists for the given learning algorithm. Proving hardness for a fixed, singleton function class requires proving hardness of the full parity, a result that has only been recently proven (Abbe et al., 2025; Shoshani & Shamir, 2025). Finally, we note that all these results are concerned with composing in-context information; extending the negative results to composing in-weights information (within the associative view) likely requires non-trivial extensions, which we point out as an open theoretical question.

## B.3 In-context graph tasks

Graph tasks have been studied extensively in the setting where each context corresponds to a unique graph. We emphasize that this is a very different setting. Indeed, a takeaway from our work is to be deliberate not to conflate insights from the in-context setting with that of the in-weights setting (a distinction that is rarely made explicit in literature).

While Bachmann & Nagarajan (2024) identify the path-star topology as a failure case for next-token learning, Frydenlund (2024; 2025) demarcate the extent of this failure in the same in-context setting, whereas Brinkmann et al. (2024) report positive path-finding results in other graph topologies. Others (Saparov et al., 2024; Sanford et al., 2024a; Yehudai et al., 2025) study Transformers in various in-context graph tasks like search and counting. Connections between in-context graph tasks and spectral biases exist (Cohen et al., 2025; Park et al., 2025a) but should not be confused with the spectral bias in in-weights tasks. While all these works study symbolic graph tasks, other works have empirically identified the limitations on graphs described in natural language (Guo et al., 2023; Wang et al., 2023; Dai et al., 2025). Kim et al. (2022); Ying et al. (2021b) propose algorithmic ideas for encoding graphs as inputs to Transformers. Finally, various failures (Momennejad et al., 2023; Dziri et al., 2024; Valmeekam et al., 2023a;b;c; Shojaee et al., 2025) have been reported on in-context tasks, including planning tasks, framed as word problems.

## B.4 Analysis of Transformer memory

**Associative memory.** The concept of associative memory dates back to theories of how information is stored in the brain (Longuet-Higgins et al., 1970; Willshaw et al., 1969). These ideas have since been explicitly modeled through various architectures such as Hopfield networks (Hopfield, 1982; Ramsauer et al., 2021), energy-based models (Krotov & Hopfield, 2016), and other modern Transformer-style inventions (Le et al., 2020; Hoover et al., 2023). Closest to us are works that analyze how architectures implicitly behave as associative memory storage, such as in autoencoders (Radhakrishnan et al., 2020) or Transformers (Bietti et al., 2023; Cabannes et al., 2024a; Nichani et al., 2024; Geva et al., 2021; Schlag et al., 2021; Sukhbaatar et al., 2019). We emphasize that this view is sufficient to understand Transformer behavior on disjoint facts, evidenced by the rich empirical literature built on this view. The geometric view only seems necessary when the facts become interdependent.

**Expressive capacity.** Theoretical works have quantified bounds on the expressive capacity of models when it comes to memorizing sequences (Mahdavi et al., 2024; Kajitsuka & Sato, 2025; Madden et al., 2025; Kajitsuka & Sato, 2024; Kim et al., 2023) as opposed to associations between pairs of bigrams. These do not comment on the learning dynamics. These works typically assume that the token embeddings are all well-separated from each other (an assumption that empirically breaks in our setting). In light of the geometric view, it is necessary to restate expressive capacity bounds in terms of geometric capacity of a network and the "geometric complexity" of the dataset.

**Empirical analyses.** Other works (Allen-Zhu & Li, 2025; Morris et al., 2025; Roberts et al., 2020; Pan et al., 2025) have performed careful empirical analyses of scaling laws for memorization, and quantified memorization in terms of "bits per parameter count", known as bit complexity (Vardi et al., 2022), which is related to our notion of bit count in Proposition 1. Zucchet et al. (2025) empirically analyze the dynamics behind how facts are memorized in a model. Others (Liu et al., 2024; Zhang et al., 2025) have proposed methodological improvements to acquiring knowledge in a Transformer; of relevance to us is the finding in Zhang et al. (2025) that training a model simultaneously on both facts and question-answering is a better way to integrate knowledge into the parameters.

**Mechanistic interpretability.** There have been mechanistic investigations into how Transformers perform fact recall (Lv et al., 2024; Geva et al., 2023) and where facts are stored in a transformer (Geva et al., 2021) and how it can be edited (Zhu et al., 2020; Meng et al., 2022). Similar attempts have been made in traditional classifier networks (Baldock et al., 2021; Maini et al., 2023; Stephenson et al., 2021). Directly related to us are the works of Khona et al. (2024) and Yao et al. (2025); Biran et al. (2024) who perform a mechanistic interpretability analysis of how multi-hop recall works.

### B.4.1 IN-CONTEXT VS. IN-WEIGHTS LEARNING

The dichotomy between drawing information from context vs. drawing information from the weights has been studied in various angles. Some have looked at this from the aspect of two competing circuits relying on one source vs. the other (Chan et al., 2022b;a; Neeman et al., 2023; Cheng et al., 2024). Others have looked at it in the context of the learning paradigms of in-context learning and finetuning the weights (Mosbach et al., 2023; Lampinen et al., 2025). Closer to us, the stark in-context vs. in-weights disparity when it comes to handling global relationships has been emphasized in Wang et al. (2024a); Geerts et al. (2025), for which our results provide further evidence.

### B.5 OTHER FOUNDATIONAL WORKS ON GENERALIZATION AND MEMORIZATION

**Spectral and simplicity bias.** The type of spectral bias we study in memorization and sequence modeling must be distinguished from the one studied in generalization and traditional classification and regression settings (Rahaman et al., 2019; Xu, 2018; Kalimeris et al., 2019; Arpit et al., 2017; Ronen et al., 2019; Bietti & Mairal, 2019). In these earlier studies, the spectrum is that of a continuous function or a decision boundary (e.g., say, the Fourier components of a polynomial), whereas the spectrum we are concerned with is of a discrete, combinatorial object, namely the graph adjacency matrix. Furthermore, the core idea of these earlier studies is that the topmost eigenvectors (the lowermost frequencies) are learned first and the rest picked up later, suggesting the need to early-stop to preserve the top eigenvectors; whereas in our setting, longer training is required to filter out the bottom eigenvectors.

**Memorization.** Our work is also orthogonal to the seminal works of Zhang et al. (2017); Neyshabur et al. (2015) who were concerned with classical generalization tasks that possess statistical redundancies. Their argument is that explicit pressures cannot explain why representations arise in such tasks, but implicit pressures may. Our memorization task on the other hand is in sequence modeling, and lacks statistical redundancies; both explicit and implicit pressures do not suffice to explain the geometric representations. Our work is also orthogonal to the foundational work of Feldman (2020) who argue that memorizing the quirks of a training set can be *necessary* for generalization in long tail datasets.

### B.6 ANALYSES OF GRAPH AND WORD EMBEDDING METHODS

Much attention has been given to characterizing what embeddings are learned by various contrastive losses such as `Node2Vec`. Most of these are on losses simpler than the softmax loss. However,

a recent line of work on next-token prediction with the softmax loss (Zhao et al., 2025; Zhao & Thrampoulidis, 2025; Thrampoulidis, 2024) studies the geometry of `Word2Vec` models. The approach here is orthogonal to ours as they make connections to a support vector machine rather than the graph spectrum. The setting also has certain technical differences under which the training dynamics turn out to be very different e.g., the model converges in direction. The difference here likely stems from the fact that in these studies the embedding and unembedding matrices are not weight-tied and correspond to different spaces (words vs. contexts).

The connection to a graph spectrum has been made in many other analyses. These analyses focus on simpler loss called the negative sampling loss where the closed form expression for the inner products is straightforward (namely, the so-called pointwise mutual information (PMI) matrix, as discovered in Levy & Goldberg (2014)). This analysis does not however tell us what exactly the embeddings are. The connection between these embeddings and the graph spectrum has been established in adjacent settings, like with DeepWalk (Qiu et al., 2018) (where the objective is explicitly multi-hop), in low-rank settings like SimCLR (HaoChen et al., 2021; Tan et al., 2024) or with quadratic losses with early stopping (Karkada et al., 2025) or the softmax loss with rank 1 (Jaffe et al., 2020). Other analyses of `Node2Vec` focus on specific types of graphs such as stochastic block models (Davison et al., 2024; Harker & Bhaskara, 2024). We refer the reader to Goyal & Ferrara (2018) for a survey of graph embedding methods. Finally, we clarify that these methods and our insights must not be confused with graph neural networks (Ying et al., 2021a; Zhou et al., 2020), where the graphs are not stored in parametric memory, but presented as input.

**Linear representation hypothesis.** A long-studied geometric concept in language models is the concept of linear representations in analogies (Park et al., 2024; 2025b; Elhage et al., 2022; Mikolov et al., 2013b). The introduction of certain concepts often takes a linear direction, surprisingly, independent of context i.e., going from `cow` to `calf` takes the same direction as `cat` to `kitten`, independent of the source in the context, (`cow`, `cat`). This linear structure, is related, but neither reducible to nor reducible from geometric memorization. Many theories have been proposed to model the linear geometry and semantics of these embeddings (Gittens et al., 2017; Allen & Hospedales, 2019; Ethayarajh et al., 2019; Allen et al., 2019; Hashimoto et al., 2016; Jiang et al., 2024b). These studies are orthogonal since their contribution lies in identifying what structures exist in word-context relationships for such geometries to arise in the embeddings. This is akin to identifying structures in the adjacency matrix, while our analysis is agnostic to such structures. There are many other complementary analyses of these embeddings, both theoretical (Grohe, 2020) and empirical (Mimno & Thompson, 2017; Chen et al., 2021; Chang et al., 2022; Li et al., 2020).

## C    DETAILED BACKGROUND ON THE PATH-STAR TASK

A path-star graph $\mathcal{G} = (V, E)$ is a special tree graph from Bachmann & Nagarajan (2024) that has a central node named $v_{\text{root}}$ with multiple paths emanating from it. One of the leaf nodes is specified as $v_{\text{goal}}$. The task is to find the unique path from $v_{\text{root}}$ to $v_{\text{goal}}$. To solve this task, one may either plan—by searching over the paths and backtracking—or execute a much simpler solution by following the unique path back from $v_{\text{goal}}$, and then outputting the reverse of this path. Although this solution is algorithmically straightforward, and although a Transformer can even be shown to learn this solution under a simple multi-token modification to the objective, the next-token objective itself fails to learn this task even with sufficient amounts of data. Thus, the failure comes from optimization under the next-token objective. In this sense, the path-star task is known to be a minimal textbook adversarial example to next-token learning.

**In-context path-star task.** In the in-context version of the task, the model is given the prefix $p = (\text{adj}(\mathcal{G}), v_{\text{root}}, v_{\text{goal}})$ that provides a randomized adjacency list, and indicates the start and goal nodes, in the model's context. The model must produce the true path as response, $r = (v_{\text{root}}, \ldots, v_{\text{goal}})$. To cast this as a learning task, we define a distribution $\mathcal{D}_{d,l}$ corresponding to graphs of the same topology, degree $d$ and path length $l$. To sample $p \sim \mathcal{D}_{d,l}$, we uniformly sample node values from a vocabulary $\mathbb{V}$ to create the graph $G$; given this, we can randomize the adjacency list $\text{adj}(\mathcal{G})$. Figures 6 and 7 contrast the two evaluation regimes we use throughout: in-weights (edge memorization & path finetuning) versus in-context (graph in the prompt).

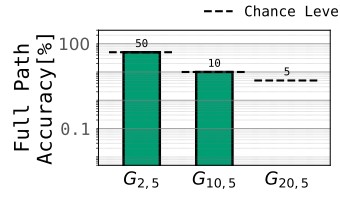
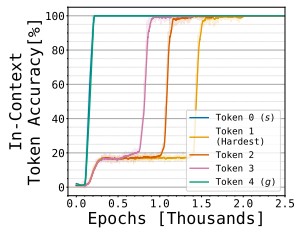
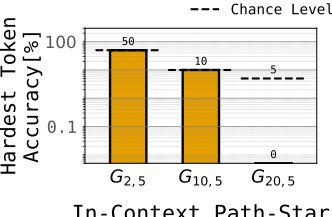

Figure 5: **Failure of Transformer in in-context path-star task.** We report the failure of next-token Transformers in the in-context version of the path-star task, reproducing results from B&N'24. **(left)** Full path accuracy remains at chance level across different small graph sizes. **(Middle)** Learning order of tokens with teacherless (multi-token-trained) objective shows a clear right-to-left learning cascade. **(right)** The hardest (first) token given the leaf fails to be learned in isolation, contrasting sharply with in-weights success shown in Fig. 2.

C.1  FAILURE OF NEXT-TOKEN LEARNING IN THE IN-CONTEXT PATH-STAR TASK

Next-token learners trained on samples from the above task are known to fail *in-distribution* i.e., even on unseen graphs of the *same* topology (with only variations in the node identities and adjacency list randomization), the model uniformly at random picks a path to follow from the center. The mechanism of failure plays out in two stages during training.

**The Clever Hans Cheat.** Early on during training, the model fits the later tokens in the target — concretely nodes $v_2, \ldots, v_{\texttt{goal}}$ — as the unique child of the previous token provided in the input during next-token training. This is known as a *Clever Hans cheat*: the model uses a local rule that relies on witnessing part of the ground truth prefix $(\boldsymbol{p}, \boldsymbol{r}_{<i})$ to predict the next token $r_i$ — as against only using the prefix $\boldsymbol{p}$ to predict the answer tokens. Arguably, this happens because the cheat is much simpler to learn — as an induction head (Olsson et al., 2022) — than the more complex solutions that rely only on $\boldsymbol{p}$ (which involves search-and-backtracking or the simpler lookahead-and-reverse). Simpler predictive features are prioritized early in training (Arpit et al., 2017; Shah et al., 2020; Rosenfeld & Risteski, 2024), which starves gradient signals from more complex features (Pezeshki et al., 2021).

**The Indecipherable Token.** Therefore, in the second stage towards failure, the model attempts to learn the first token $r_i$—the key decision-making token—purely based on information about the graph in the prefix $\boldsymbol{p}$, and without any gradient signal about the later tokens. This is however a computationally hard "needle-in-the-haystack problem" (Wies et al., 2023): the model needs to find a complex end-to-end algorithm with $\ell$ subroutines in it, without any intermediate supervision for those subroutines. The first token thus becomes "indecipherable" and the model simply memorizes it on the training data; during test-time the model subsequently predicts the first token as a random neighbor, and then continues following down the path by applying the local follow-the-child rule it learned through the Clever Hans cheat.

Fig. 5 demonstrates these failure modes empirically, showing that next-token prediction achieves only chance-level performance on path-star graphs of varying sizes, with the first token remaining particularly difficult to learn in isolation.

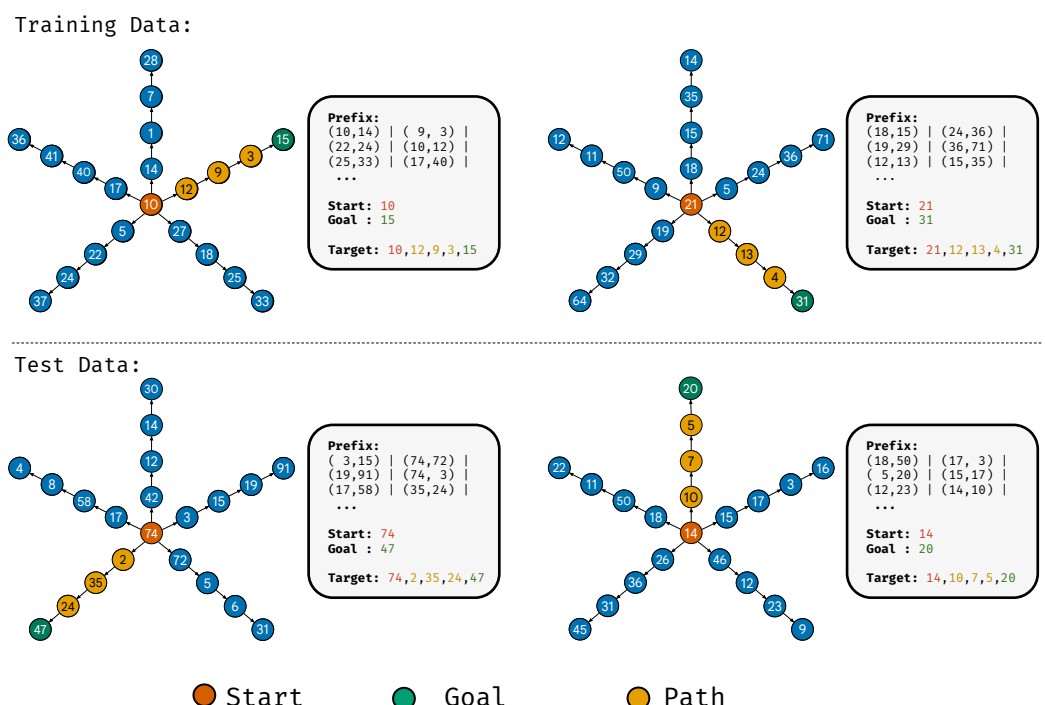

Figure 6: **Overview of in-context path-star task of B&N'24.** Each training and test example corresponds to a fresh, randomly-labeled path-star graph (a tree graph where only the root node branches into $d$ paths of length $\ell$). For each example, the prefix specifies a randomized adjacency list (of edge bigrams) of the corresponding graph, followed by $(v_{\text{root}}, v_{\text{goal}})$. The target is the full path $(v_{\text{root}} \rightarrow v_{\text{goal}})$ in that graph.

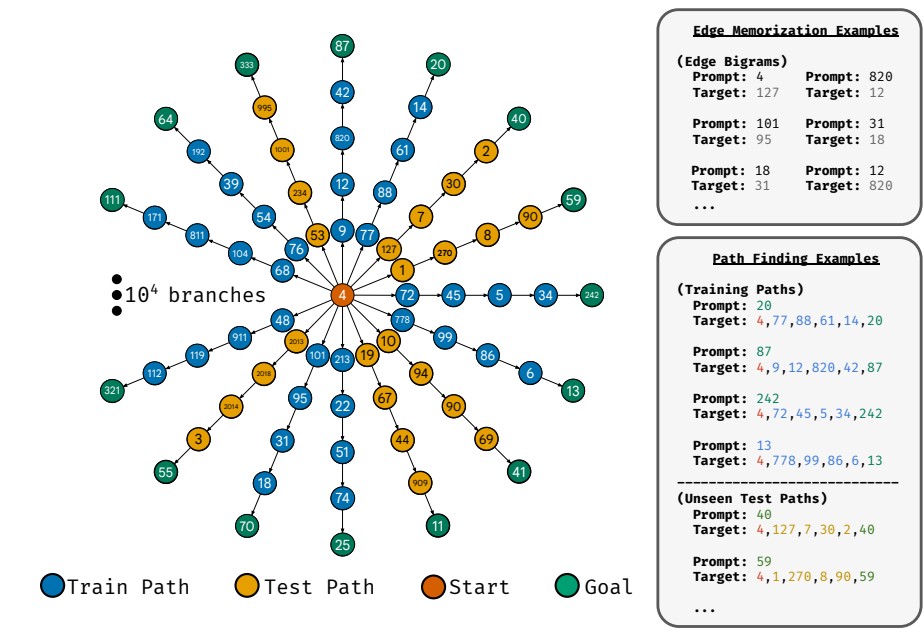

Figure 7: **Overview of our in-*weights* path-star task.** All examples are derived from a fixed path-star graph. Training involves two types of examples: (i) **edge memorization** examples (ii) **path-finding** examples, where the prefix is some leaf, and the target is the full path. Test examples are path examples corresponding to a held-out set of leaves.

# D  EXPERIMENTAL SETUP

## D.1  GRAPHS, TOKENS, AND DATA CONSTRUCTION

**Path-star graphs.** We denote by $\mathcal{G}_{d,\ell}$ a path-star graph with central node $v_{\texttt{root}}$, degree $d$ (number of arms), and path length $\ell$ per arm. The $i$-th arm consists of the node sequence $(v_{\texttt{root}} = v_0^{(i)}, v_1^{(i)}, \ldots, v_{\ell-1}^{(i)})$ where the last node in the sequence is $v_{\ell-1}^{(i)} = v_{\texttt{leaf}}^{(i)}$.

**Vocabulary.** Each node is represented by a unique numerical token. We reserve several special tokens: [PAUSE] (a compute/pause token), [PAD] (a padding token), optional directional tokens (>, <), and task-specific prefix tokens ([EDGE], [PATH]). The effective vocabulary size is $|\mathbb{V}| = |\text{nodes}| + 9$.

**Edge-memorization datasets for local supervision.** We form directed bigrams $(u, v)$ when $v$ is a child of $u$ to define a distribution $\mathcal{D}_{\texttt{edge}}^{\rightarrow}$. Conversely, we use examples of the form $(v, u)$ where $u$ is the parent of $v$ to define $\mathcal{D}_{\texttt{edge}}^{\leftarrow}$; and $\mathcal{D}_{\texttt{edge}}$ is their union. Training sequences of edge bigrams are simple two-token sequences "$u\ v$" sampled uniformly. All these examples, be it forward or backward, provide only local supervision.

**Path-finding datasets.** We define a path distribution $\mathcal{D}_{\texttt{path}}^{\rightarrow}$, where an example is the pair $(\boldsymbol{p}, \boldsymbol{r})$ with $\boldsymbol{p} = (v_{\texttt{leaf}}^{(i)}, \overbrace{[\texttt{PAUSE}], \ldots, [\texttt{PAUSE}]}^{P \text{ tokens}})$ and $\boldsymbol{r} = (v_{\texttt{root}}, v_1^{(i)}, \ldots, v_{\ell-1}^{(i)})$. The leaf node is sampled uniformly. We finetune on a subset of leaves and *evaluate on held-out leaves*. Unless stated otherwise, decoding is greedy (top-1). The arrow in $\mathcal{D}_{\texttt{path}}^{\rightarrow}$ denotes that this is the *forward* path; we also experiment with predicting the reverse goal-to-start path, denoted by the distribution $\mathcal{D}_{\texttt{path}}^{\leftarrow}$.

**In-context datasets.** Adapted from Bachmann & Nagarajan (2024), the prefix contains a randomized adjacency serialization and $(v_{\texttt{root}}, v_{\texttt{goal}})$; the target is the full path. Where NTP fails in-context, we use the teacherless objective of Bachmann & Nagarajan (2024) for comparison plots (details in §D.3).

## D.2  MODEL ARCHITECTURE

### D.2.1  IN-WEIGHTS PATH-STAR TASK EXPERIMENTS

**Backbone.** The main experiments appearing in §2 use a decoder-only Transformer (GPT-mid) with a causal mask, pre-norm LayerNorm, sinusoidal positional embeddings (Vaswani et al., 2017), GELU MLPs, and tied input/output token embeddings. Additionally, experiments in §E.1 employ a Mamba sequence model (Gu & Dao, 2023) of comparable scale.

**Default Transformer configuration.**

- Layers $N_{\texttt{layer}} = 12$, model width $m_{\texttt{width}} = 384$, heads $m_{\texttt{head}} = 8$.
- Dropout 0 on attention and MLP blocks (synthetic setting), label smoothing 0.
- Context length set to accommodate the longest training sequence (edges: 2; paths: $\ell + 1 + N_{\texttt{pause}}$) with margin.

**Mamba configuration.** The Mamba models in §E.1 use an equivalent depth and hidden dimension to the Transformer baseline. The sequence model parameters include the state dimension $d_{\text{state}} = 16$, convolution kernel size $d_{\text{conv}} = 4$, and expansion factor expand $= 2$, which follow the standard values from the original Mamba implementation.

**Embedding layer.** Token embeddings are stored in $\mathbf{V} \in \mathbb{R}^{|\mathbb{V}| \times m_{\texttt{width}}}$, with output projection weights tied to $\mathbf{V}$.

**Variants.** For larger graphs $\mathcal{G}_{10^4, 10}$, we scaled $m_{\texttt{width}}$ proportionally to 784 in our hyperparameter grid search. For the Mamba architecture, we also varied the expansion factor to 4 and the state dimension to 32.

### D.2.2  TINY MODEL ARCHITECTURES

For quicker toy experiments on small-scale graphs (§E.3, §F.1.2), including the *Tiny Path–Star*, *Tiny Grid*, *Tiny Cycle*, and *Tiny Irregular* graphs, we used reduced-size architectures.

**Models.** We evaluated three model types: (1) a Transformer (TinyGPT), (2) a feed-forward neural network with the same configuration but without attention, and (3) a Mamba model of comparable scale to (1).

**Configuration.** For the *Tiny Path–Star*, *Tiny Grid*, and *Tiny Cycle* graphs, all models used a single layer ($N_{\texttt{layer}} = 1$), embedding dimension $m_{\texttt{width}} = 32$, and $m_{\texttt{head}} = 8$. The Mamba variant further included $d_{\texttt{state}} = 8$, $d_{\texttt{conv}} = 4$, and expand $= 2$.

For the *Tiny Irregular* graphs, for all models, we used $N_{\texttt{layer}} = 3$, $m_{\texttt{width}} = 256$, and the Mamba parameters $d_{\texttt{state}} = 8$, $d_{\texttt{conv}} = 4$, expand $= 4$. We use a larger number of layers since we found that otherwise the model does not achieve perfect edge-memorization with frozen embeddings.

**Additional details.** In associative-memory settings (e.g., left-column visualizations in Figure 1), token embeddings were frozen. All models employed weight tying between input and output embeddings.

## D.3 TRAINING AND OPTIMIZATION

**Objective.** We use next-token cross-entropy over the causal prefix. For first-token-only experiments, the loss is restricted to the first target position. Although we train all our models to $50,000$ epochs, we *only need about a couple of thousand epochs* to see accuracy gains (e.g., see the per-token accuracy plots in Fig. 2).

**Optimizer and schedule.** We use the AdamW optimizer with a weight decay of $0.01$. The learning rate follows a cosine decay schedule with a linear warm-up. In the two-phased edge-memorization ablation experiment of §F.3, the peak learning rate for edge memorization (Phase 1) is $1 \times 10^{-2}$ and for path finetuning (Phase 2) is $5 \times 10^{-5}$.

**Batching.** We used a range of different batch sizes of $\{64, 128, 256, 512, 1024\}$.

**PAUSE tokens.** We append $N_{\texttt{pause}} \in \{0, 2, 4, 6, 10\}$ pauses in $\mathcal{D}^{\rightarrow}_{\texttt{path}}$ to provide compute budget (no labels on pause positions). The chosen $N_{\texttt{pause}}$ for each $\mathcal{G}_{d,\ell}$ is given in Table 2.

Table 2: Default hyperparameters by graph size. Values denote the settings used.

| Graph $\mathcal{G}_{d,\ell}$ | $N_{\texttt{layer}}$ | $m_{\texttt{width}}$ | $m_{\texttt{head}}$ | $N_{\texttt{pause}}$ | Peak LR |
|---|---|---|---|---|---|
| $\mathcal{G}_{5\times10^3,5}$ (In-Weights) | 12 | 384 | 8 | 5 | $5 \times 10^{-5}$ |
| $\mathcal{G}_{10^4,6}$ (In-Weights) | 12 | 384 | 8 | 6 | $5 \times 10^{-5}$ |
| $\mathcal{G}_{10^4,10}$ (In-Weights) | 12 | 784 | 8 | 10 | $5 \times 10^{-5}$ |
| $\mathcal{G}_{2,5}$ (In-Context) | 12 | 384 | 8 | n/a | $1 \times 10^{-4}$ |
| $\mathcal{G}_{10,5}$ (In-Context) | 12 | 384 | 8 | n/a | $1 \times 10^{-4}$ |
| $\mathcal{G}_{20,5}$ (In-Context) | 12 | 784 | 8 | n/a | $1 \times 10^{-4}$ |

## D.4 EVALUATION PROTOCOLS AND METRICS

**Forward vs. reverse.** We evaluate forward generation ($v_{\texttt{root}} \rightarrow v_{\texttt{leaf}}$) and reverse generation ($v_{\texttt{leaf}} \rightarrow v_{\texttt{root}}$). Reverse is algorithmically trivial after edge memorization; forward is the non-trivial case we care about.

**Metrics.**

- *Exact-match path accuracy* / *Full path accuracy*: fraction of held-out leaves whose entire path is generated correctly.
- *First-token accuracy* / *Hardest token accuracy*: accuracy of the first hop (hardest token) given the leaf.
- *Per-token accuracy over epochs* / *Token accuracy*: accuracy at each target position, tracked through training (used in Fig. 24).

**Baselines.** Random choice among $d$ branches gives $1/d$ first-token accuracy and near-zero exact-match.

## D.5 IMPLEMENTATION AND COMPUTE

Code is implemented in `PyTorch` with standard Transformer components. All runs fit on a single modern GPU (e.g., `A100-40GB`); per-figure training time depends on $(d, \ell)$ and batch size.

# E  EXPERIMENTS ON BROADER SETTINGS

In this section, we demonstrate that our findings generalize to various other deep sequence architectures and various other large and smaller graphs.

## E.1  PATH-STAR TASK ON MAMBA SSM

This section demonstrates that the implicit reasoning on path-star graphs observed for Transformers in §2.1, generalizes to the Mamba SSM architecture Gu & Dao (2023) too. Fig. 8 is the counterpart of the Transformer accuracy plots in Fig. 2; Fig. 9 the counterpart to Transformer heatmaps in Fig. 16; Fig. 10 the counterpart to the UMAP topology of Transformers in Fig. 18. A notable difference here is that the Mamba model presents a strong geometry even when trained only on the edges (as seen in Fig. 9c).

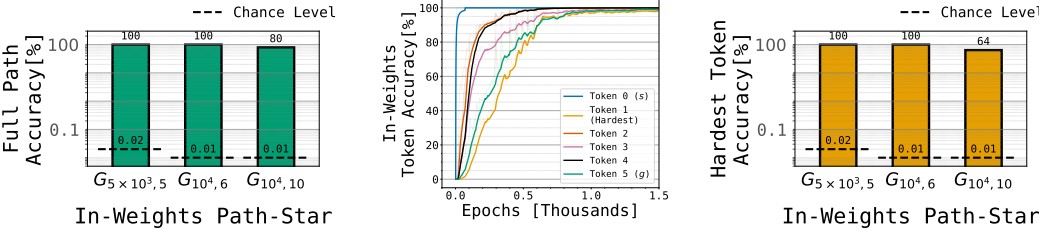

Figure 8: **(left) Success of in-weights path-star task for *Mamba*.** This figure is a counterpart to Fig. 2. A next-token-trained Mamba achieves perfect or highly non-trivial accuracy on large path-star graphs $\mathcal{G}_{d,\ell}$ (Observation 1a). **(middle) Learning order of tokens.** The tokens of a path are not learned in the reverse order i.e., the model does not learn the right-to-left solution. Thus, gradients from the future tokens are not critical for success (Observation 1b). **(right) Success of hardest-token-only task.** In fact, the hardest token (the first token) given the leaf is learned in isolation to non-trivial accuracy (Observation 1c). Success of this $\ell$-fold composition task is hard to explain within the associative memory (§2.2).

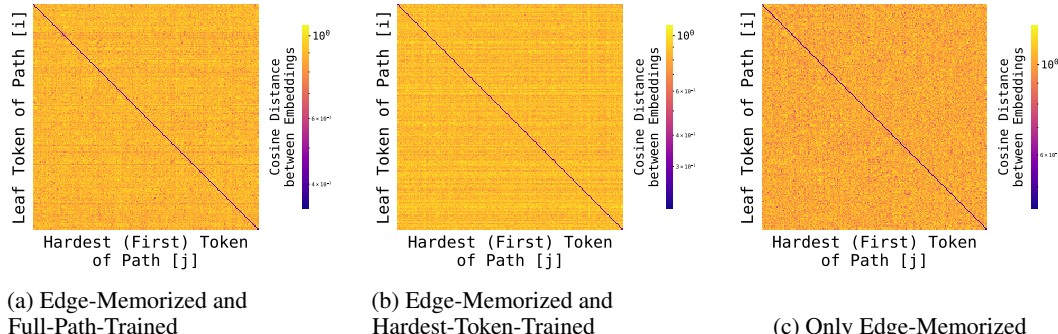

(a) Edge-Memorized and Full-Path-Trained

(b) Edge-Memorized and Hardest-Token-Trained

(c) Only Edge-Memorized

Figure 9: **Evidence of global geometry in path-star task for *Mamba*.** This figure is a counterpart to Figs. 3 and 16 for the Mamba SSM architecture. Recall that entry $(i, j)$ is the mean cosine distance between the leaf token in (an unseen) path $i$ (row) and first/hardest token on (an unseen) path $j$ (col). Each heatmap corresponds to a different training objective: Left: trained on edges and path-finding task ($\mathcal{D}_{\text{edge}} \cup \mathcal{D}_{\text{path}}^{\rightarrow}$); Middle: trained on edges and hardest-token-finding task (not presented in the main paper); Right: edges only ($\mathcal{D}_{\text{edge}}$). We find that even on these unseen paths, the leaf and first token embeddings cluster together, regardless of whether path-finding supervision exists. Interestingly, compared to the Transformer in Fig. 16, the Mamba trained only on local supervision (right) exhibits a much stronger geometry.

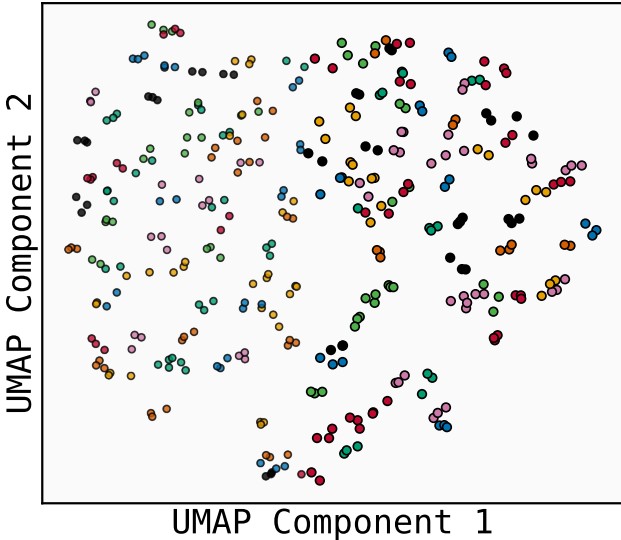

Figure 10: `UMAP` **projection of token embeddings of *Mamba* exhibits path-star topology.** We corroborate the `UMAP` (McInnes et al., 2018) observations from the Transformer (Fig. 18) in Mamba SSM here. Each point is a node embedding; color indicates path identity. Different paths form separated clusters (although this clustering is weaker than for a Transformer); the central node $v_{\texttt{root}}$ is excluded since it is shared across all paths. The graph has degree $10^4$ and path length 6 (including the root). Axes are `UMAP` components (arbitrary units). Note that we have re-used each color for multiple paths.

## E.2 OTHER LARGE, HARDER PATH-FINDING GRAPHS

Next, we consider variants of the path-finding graph. While the path-star graph is adversarially constructed in certain ways, there is only one decision-making step, which makes planning simpler in a certain way. To make the task harder, we introduce a branching at every node along each path. (Similar tree variants have been considered in Frydenlund (2025) for the in-context task, but we are interested in in-weights tasks.)

In the *tree*-star graph, $\mathcal{T}_{d,\ell}$, there is a central node with degree $d$ and each child node except the leaf has a fixed degree of 2, as visualized in Fig. 11a. The path-length from the root to any leaf is $\ell$. In this graph, two types of learning tasks can be considered, depending on how we split the test and training paths. For a no-overlap setting, we could sample all training paths from one subset of trees, and the test paths from the remaining trees; we call this the *split at first token* setting, since the test/train split is determined by the first token. A second setting—-with some test-train overlap—is one where we reserve some leaves as training goals, and the rest as test-goals. Here, on any test path, a prefix may have participated in a training path. We call this the *split at leaf* setting. Note that in both variants, all nodes will be sampled as part of the edge-memorization task.

In Fig. 11b, we find that on both variants the Transformer achieves non-trivial path-finding accuracy, generalizing our results beyond the path-star task. However, we note that the test-train split at the first token is much harder to succeed at, likely due to no overlaps.

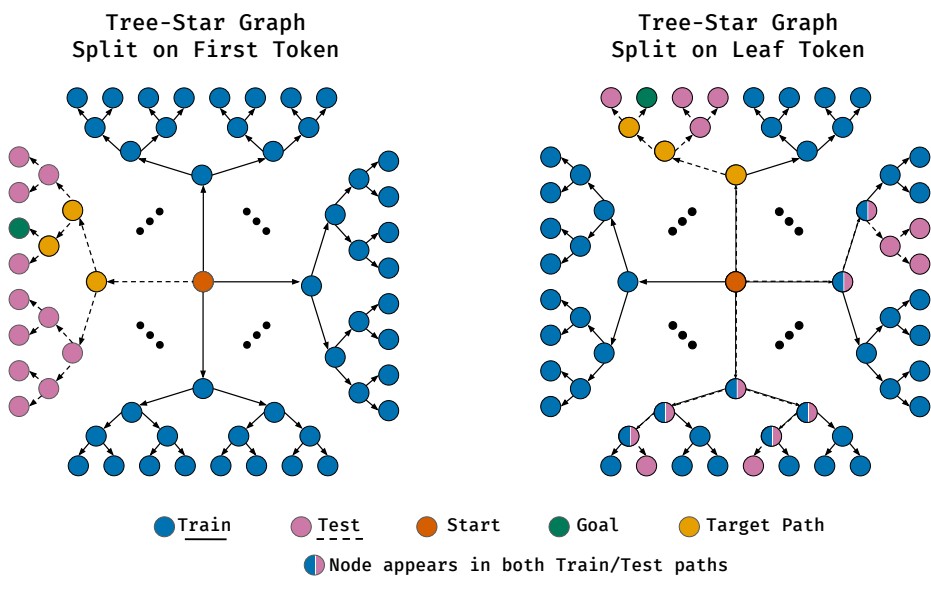

(a) In-Weights Tree-Star

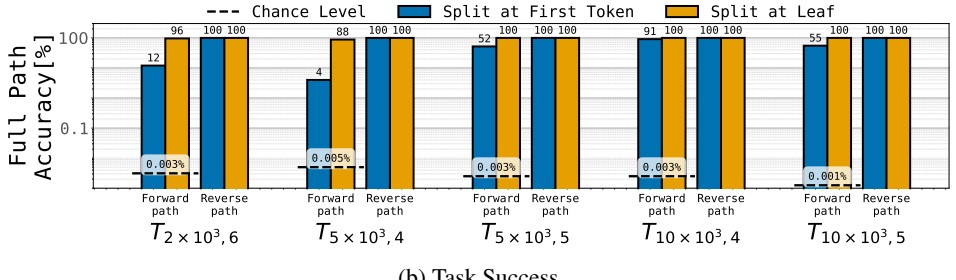

(b) Task Success

Figure 11: **Transformer achieves non-trivial accuracy on the harder in-weights tree-star task.** The tree-star task of §E.2 introduces decision-making at every step of the path, not just the first token. There are two variants of this task based on the test-train split. In the *split on first token* variant (**top-left**), we reserve some of the trees for generating training paths, and the rest for test paths. In the *split on leaf token* variant (**top-right**), we reserve some leaves for training and the rest for testing; some nodes may be sampled for path-finding during both test and train time. In both tasks, we have visualized a single target path. In the **bottom** figure, we report non-trivial path-finding accuracies on both tasks, above random chance defined as $1/\texttt{num\_leaves}$.

### E.3 Tiny graphs

Besides the large path-star graph (of §2) and tree-star graphs (of E.2), we also report the embeddings learned on four tinier graphs for various architectures (details in D.2.2); the figures here are an extension of the Transformer embeddings in Fig. 1. In these experiments, we train the models purely on local supervision (the edges, presented in both directions) to $100\%$ edge-memorization accuracy—for each vertex, we ensure that its $d$ neighbors appear in the top $d$ softmax probabilities. The graphs include (a) a tiny path-star graph with four paths of length $4$, (b) a $4 \times 4$ grid graph, (c) a 15-node cycle graph and (d) an irregular graph with two asymmetryic components. Each visualization includes the embeddings from: an associative memory model (implemented with a neural network with only one trainable matrix sandwiched between (un)embedding layers), a `Node2Vec` model, the eigenvectors of the graph Laplacian, a Transformer's token embeddings, a neural network's first layer, and a Mamba SSM's token embeddings. For the `Node2Vec` model we use the top eigenvectors, and for the rest we use `UMAP` (McInnes et al., 2018) to choose the top directions.

We consolidate the various observations from these figures (Figs. 12 to 15) here:

**Observation 5.** *In the tiny graphs of Figs. 12 to 15, on various architectures (`Node2Vec`, graph spectrum, Transformer, neural network, Mamba SSM), we find that:*

1. *A geometry arises in all these architectures even without global supervision from a path-finding task (Observation 3b).*

2. *The global information in these geometries can be traced back to the eigenvectors of the graph spectrum (§4).*

3. *The geometry arises in all three deep sequence models (Transformer, Mamba SSM, neural network) even though these models can learn the data associatively using the same learning setup, with just the (un)embedding matrices frozen (Observation 3a).*

4. *The geometry of the `Node2Vec` model (which precludes associative memory), is much stronger than the deep sequence models, suggesting that the deep sequence models may be adulterated with associative memory (Hyp. 3).*

5. *We note that similar geometries arise even with only one direction presented; see §F.1.2. This is the scenario where both types of storage have the same bit and $\ell_2$ norm complexity; thus, there are no straightforward implicit pressures that encourage the geometry (Proposition 1).*

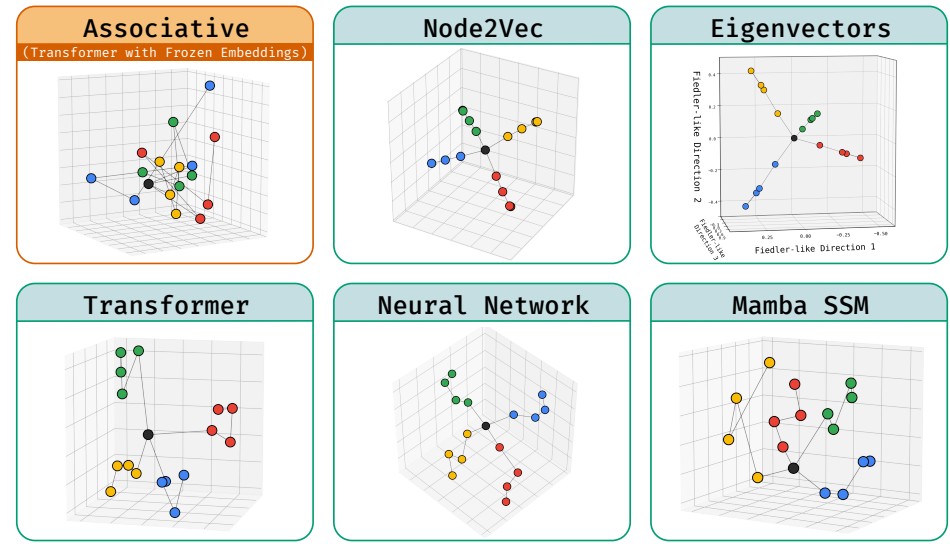

Figure 12: **Tiny path-star**: Geometries of various architectures on a smaller version of the path-star graph. See Observation 5.

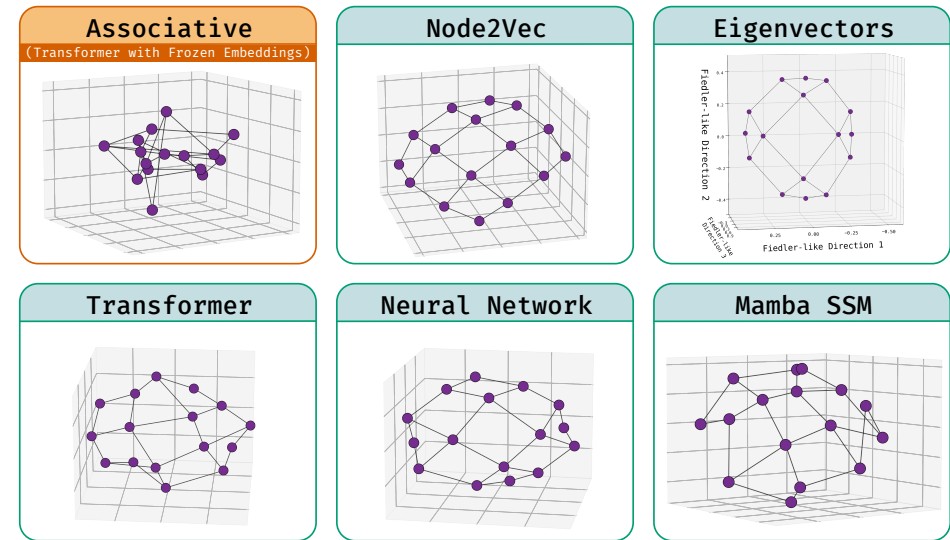

Figure 13: **Tiny grid**: Geometries of various architectures on a small $4 \times 4$ grid graph. See Observation 5.

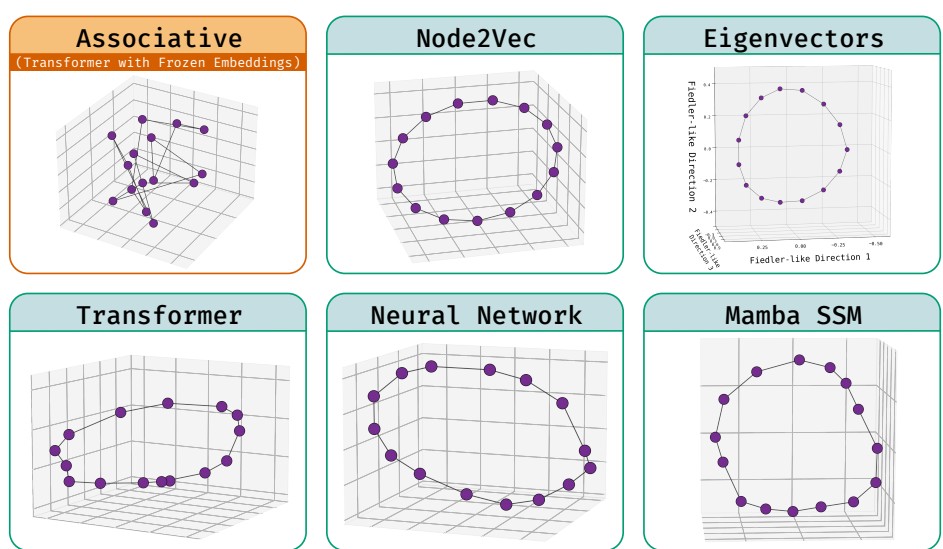

Figure 14: **Tiny cycle**: Geometries of various architectures on a small cycle graph. See Observation 5.

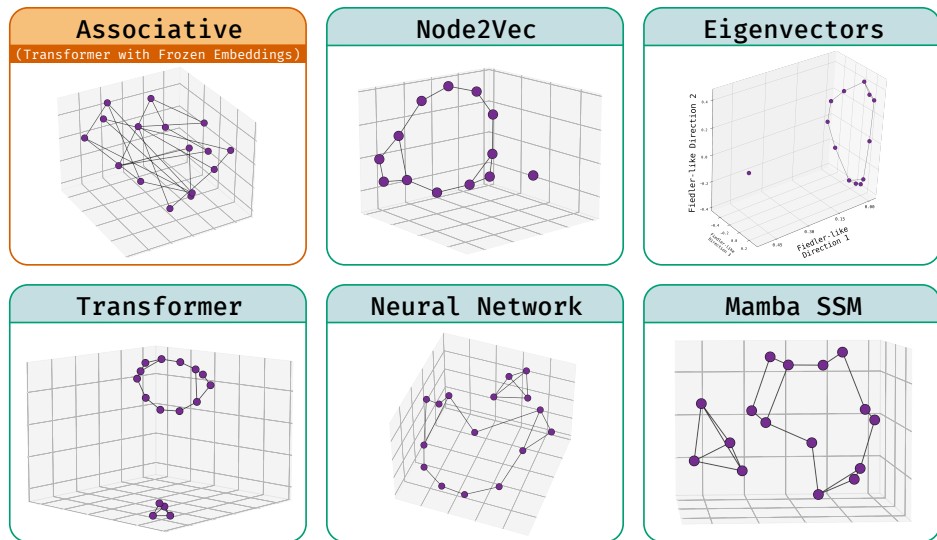

Figure 15: **Tiny irregular graph**: Geometries of various architectures on a small irregular graph of two connected components, both asymmetric. See Observation 5. Note that unlike in the other graphs, we do not use a 1-layer model here, but a 3-layered one.

In the main paper, we present heatmaps showcasing the distance between the leaf nodes and the first node in every path. We consolidate these below and also provide an additional path-to-path distance heatmap.

**Leaf–first-hop distance (Fig. 16).** In Fig. 16 (an extended version of the heatmaps in Fig. 3), entry $(i, j)$ is the cosine distance between the *leaf* embedding of path $i$ and the *first-hop (indecipherable token)* embedding of path $j$. Diagonal entries are low (a leaf lies close to its correct first hop), while off-diagonals are higher. This structure explains why the first-token-only objective succeeds in-weights (Fig. 2-(Right)): the $k$-fold composition map reduces to a local geometric step in the learned representation.

**Path–by–path distance (Fig. 17).** Instead of only analyzing the distance between the leaf and first nodes, we analyze the distance across all pairs of nodes in a path. For each pair of paths $(i, j)$, we compute the mean distance between *all* node embeddings on path $i$ and all node embeddings on path $j$.

- **Diagonal — Intra-Path Distance ($i = j$):** This value is the average distance between all unique pairs of distinct nodes within a single path. It measures how tightly clustered the path's nodes are. A smaller value indicates higher cohesion.

$$D_{i,i} = \frac{1}{\binom{\ell}{2}} \sum_{1 \le k < m < \ell} d(v_k^{(i)}, v_m^{(i)})$$

- **Off-diagonal — Inter-Path Distance ($i \ne j$):** This value is the average distance between all nodes of one path and all nodes of another. It measures how separated two distinct paths are. A larger value indicates greater separation.

$$D_{i,j} = \frac{1}{\ell^2} \sum_{k=1}^{\ell-1} \sum_{m=1}^{\ell-1} d(v_k^{(i)}, v_m^{(j)})$$

We find a similar clustering of nodes within paths here, although the diagonal is generally less vivid. For the model trained only on edge-memorization, we do not see the diagonal at all.

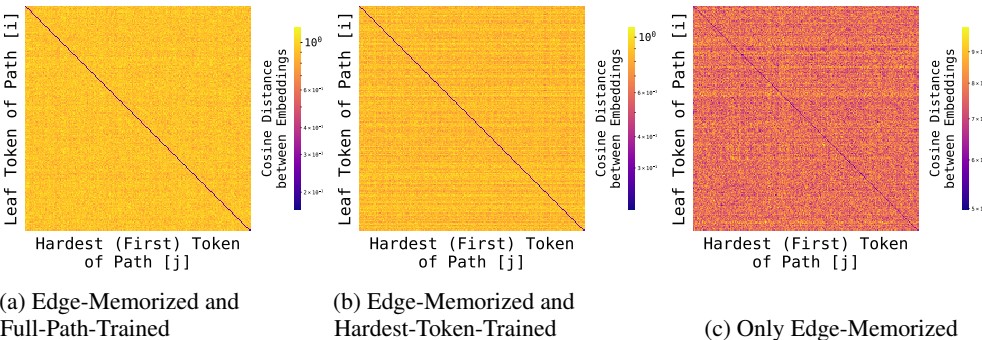

(a) Edge-Memorized and Full-Path-Trained

(b) Edge-Memorized and Hardest-Token-Trained

(c) Only Edge-Memorized

Figure 16: **Evidence of global geometry in path-star task: Leaf-first-token cosine distance between node embeddings.** We present again the heatmaps from Fig. 3 with an additional heatmap in the middle. Entry $(i, j)$ is the mean cosine distance between the leaf token in (an unseen) path $i$ (row) and first/hardest token on (an unseen) path $j$ (col). Each heatmap corresponds to a different training objective: Left: trained on edges and path-finding task ($\mathcal{D}_{\text{edge}} \cup \mathcal{D}_{\text{path}}^{\rightarrow}$); Middle: trained on edges and hardest-token-finding task (not presented in the main paper); Right: edges only ($\mathcal{D}_{\text{edge}}$). We find that even on these unseen paths, the leaf and first token embeddings cluster together, regardless of whether path-finding supervision exists; however, the geometry is strongest with global supervision (however, in Mamba SSM, even the locally supervised model shows strong geometry; see Fig. 9).

UMAP **projection (Fig. 18, zoomed in version of Fig. 3, middle).** A different way to establish geometry is to directly visualize the paths. As done in Fig. 3, we do this by projecting the token

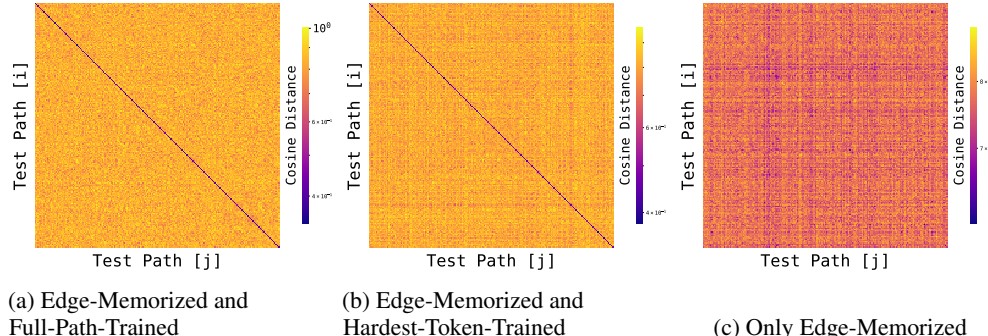

(a) Edge-Memorized and Full-Path-Trained

(b) Edge-Memorized and Hardest-Token-Trained

(c) Only Edge-Memorized

Figure 17: **Evidence of global geometry in path-star task: Pathwise average** *cosine* **distance between node embeddings.** While in Fig. 16, we reported the cosine distance between the embeddings of the *terminal* nodes of a pair of (unseen) paths, here we consider an average over all nodes in those (unseen) paths. In particular, entry $(i, j)$ is the mean *cosine* distance between nodes on path $i$ (row) and nodes on path $j$ (col). On the first two settings, as before, we find that the diagonal cells are low, implying closer embeddings. Thus, a geometry has emerged on unseen paths. However, unlike in Fig. 3, we do not see any such signal when trained only on the edge memorization task.

embeddings of all nodes with default `UMAP` settings (`neighbors=15, min_dist=0.1`). A zoomed in version of this is presented in Fig. 18. We exclude the root token embedding in these projections since that is common to all paths. We find that different paths form well-separated clusters, and within each path cluster, nodes tend to arrange from leaf toward $v_{\text{root}}$.

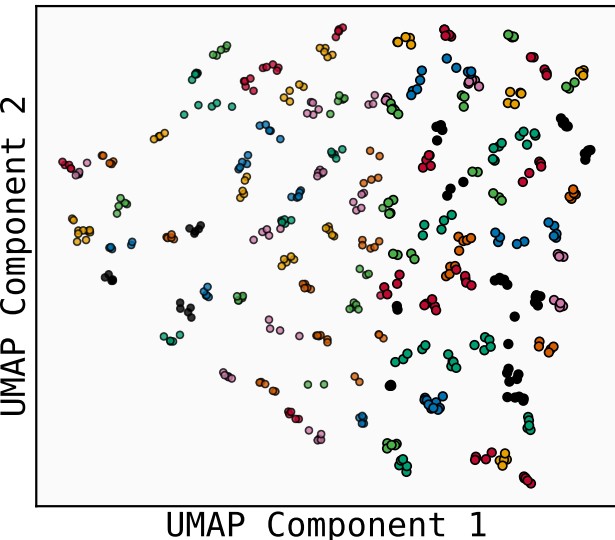

Figure 18: Zoomed in version of Fig. 3: `UMAP` **projection of token embeddings exhibits path-star topology.** Each point is a node embedding; color indicates path identity. Different paths form separated clusters; the central node $v_{\text{root}}$ is excluded since it is shared across all paths. The graph has degree $10^4$ and path length 6 (which includes the root). Axes are `UMAP` components (arbitrary units). Note that we have re-used the same color for multiple paths.

## F EDGE SUPERVISION AND TRAINING DYNAMICS

There are tangential aspects of our training that are worth elaborating on: the role of reverse edges (§F.1), the role of pause tokens (§F.2), and the role of interleaving edge-memorization (§F.3).

### F.1 THE ROLE OF REVERSE EDGES

Reverse edges seem to play a nuanced role in our observations. On the large path-star task in §2, we find it necessary to augment training on the reverse edges; on the other hand, for the tiny graphs, a geometry arises even without these reverse edges. Perhaps, reverse edges are needed for larger tasks; or perhaps, they are necessary to perform implicit reasoning and retrieval. We leave it for future work to gain greater clarity on this effect, which is tied to the reversal curse (Berglund et al., 2024; Allen-Zhu & Li, 2023).

### F.1.1 THE CRITICAL ROLE OF REVERSE EDGES IN THE LARGE PATH-STAR TASK

**Edge supervision regimes.** We evaluate three edge supervision regimes for the fixed in-weights graph: (i) *forward-only* edges $\mathcal{D}_{\text{edge}}^{\rightarrow}$, (ii) *backward-only* edges $\mathcal{D}_{\text{edge}}^{\leftarrow}$, and (iii) their mixture $\mathcal{D}_{\text{edge}} = \mathcal{D}_{\text{edge}}^{\rightarrow} \cup \mathcal{D}_{\text{edge}}^{\leftarrow}$, each combined with path supervision.

We consider two types of path-finding tasks. The first, as discussed in Section 2.1, is a *forward generation* ($v_{\text{root}} \rightarrow v_{\text{leaf}}$) task defined by $\mathcal{D}_{\text{path}}^{\rightarrow}$. Another task is *reverse generation* ($v_{\text{leaf}} \rightarrow v_{\text{root}}$), denoted by $\mathcal{D}_{\text{path}}^{\leftarrow}$. Forward path generation is non-trivial to learn as it involves planning or look-ahead, and is adversarial towards next-token learning; the reverse path however is trivial to learn on path-star graphs because each node has a unique predecessor along the target path. We must also clarify that the presence of reverse edges in itself *does not trivialize* the forward path-finding task—these edges provide only local information; thus, the success of the global path-finding task is still *non-trivial*.

We enumerate our observations from these various edge-supervision regimes below:

**Observation 6.** *(Role of reverse edges) We find in Fig. 19 that:*

1. *A Transformer trained on only the forward edges, struggles on both forward and reverse path-finding tasks (see the middle color in Fig. 19).*

2. *A Transformer trained on only the reverse edges, achieves non-trivial accuracy on the reverse path-finding task; however, it fails on the forward path-finding task (see the third color in Fig. 19).*

We suspect that the lack of reverse edges either hurts the geometry *or* hurts the retrieval ability of the model. On the other hand, the success of the reverse path-finding task with reverse-only edge memorization could be explained by the fact that the task requires no planning, as discussed in the remark below.

**Remark 1.** *We note that the asymmetry between forward and reversed path-generation tasks stems from their algorithmic complexity. The reversed path generation is algorithmically trivial on path-star graphs because each node has a unique predecessor along any target path—the model simply needs to follow the unique backward edges. Forward generation, however, requires planning (examine each outgoing path) or lookahead (track the reverse path without explicit chain-of-thought and reverse it). Indeed, for the in-context task of B&N'24, the model fails on the forward task, but strikingly succeeds on the reverse task, as corroborated in Fig. 20 (right). Even in the in-weights setting Fig. 20 (left), the reverse path is generally quicker to learn and yields higher accuracy.*

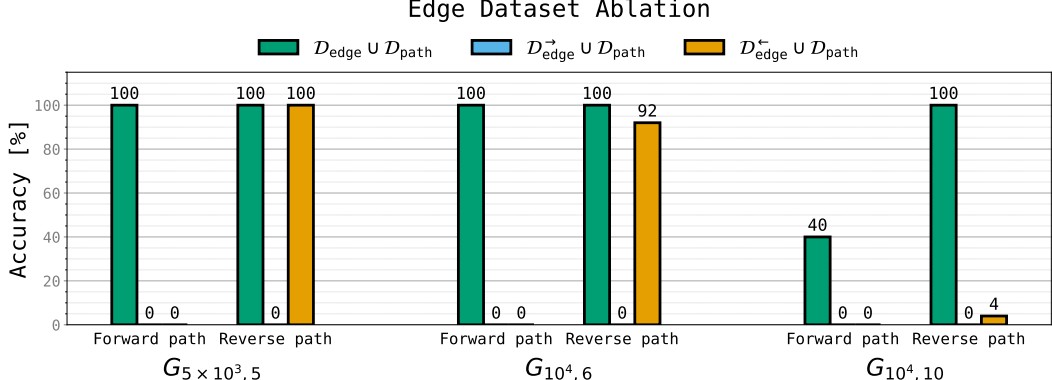

Figure 19: **Mixed edge supervision enables forward path generation while forward-only fails due to reversal curse.** Exact-match accuracy on held-out leaves for multiple path-star graphs (varying degree $d$ and path length $\ell$). As established, training on *mixed* edges $\mathcal{D}_{\mathsf{edge}}$ yields high non-trivial *forward* accuracy across graphs. But training on *forward-only* $\mathcal{D}_{\mathsf{edge}}^{\rightarrow}$ fails on both the forward and reverse tasks. This is indicative of the reversal curse. With *backward-only* edges ($\mathcal{D}_{\mathsf{edge}}^{\leftarrow}$) the model attains high accuracy primarily on *reverse* path generation for smaller graphs, This can be reconciled by noting that generating the reverse path is an easier retrieval task. Random forward path accuracy is $1/d$.

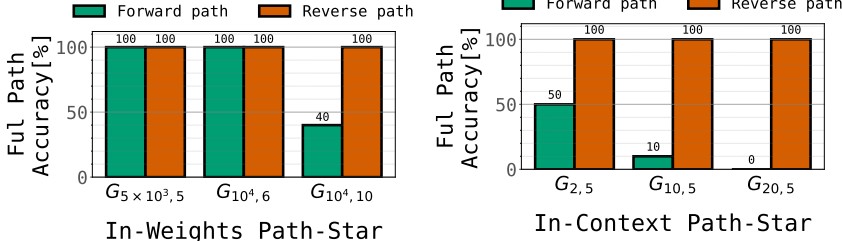

Figure 20: **Forward vs. reverse path generation:** The figure contrasts the model's performance on forward (start→leaf) and reverse (leaf→start) path generation tasks for path-star graphs learned either **in-weights** (left) or **in-context** (right). While both methods achieve perfect accuracy on the algorithmically simple reverse path task, their performance on the forward task differs dramatically. **(left)** The in-weights model succeeds at the forward task, which requires planning and look-ahead, demonstrating high accuracy even on large graphs with thousands of nodes. **(right)** In contrast, the in-context model completely fails at forward path generation. This stark difference highlights the superior capability of in-weights learning to internalize and utilize complex graph structures.

### F.1.2    TINY GRAPHS WITH UNI-DIRECTIONAL EDGE-MEMORIZATION

In Fig. 21, we revisit our tiny graphs in §E.3, and examine the embeddings of a Transformer when it memorizes only one direction of the edges. We find that a geometry still arises, although a bit weaker.

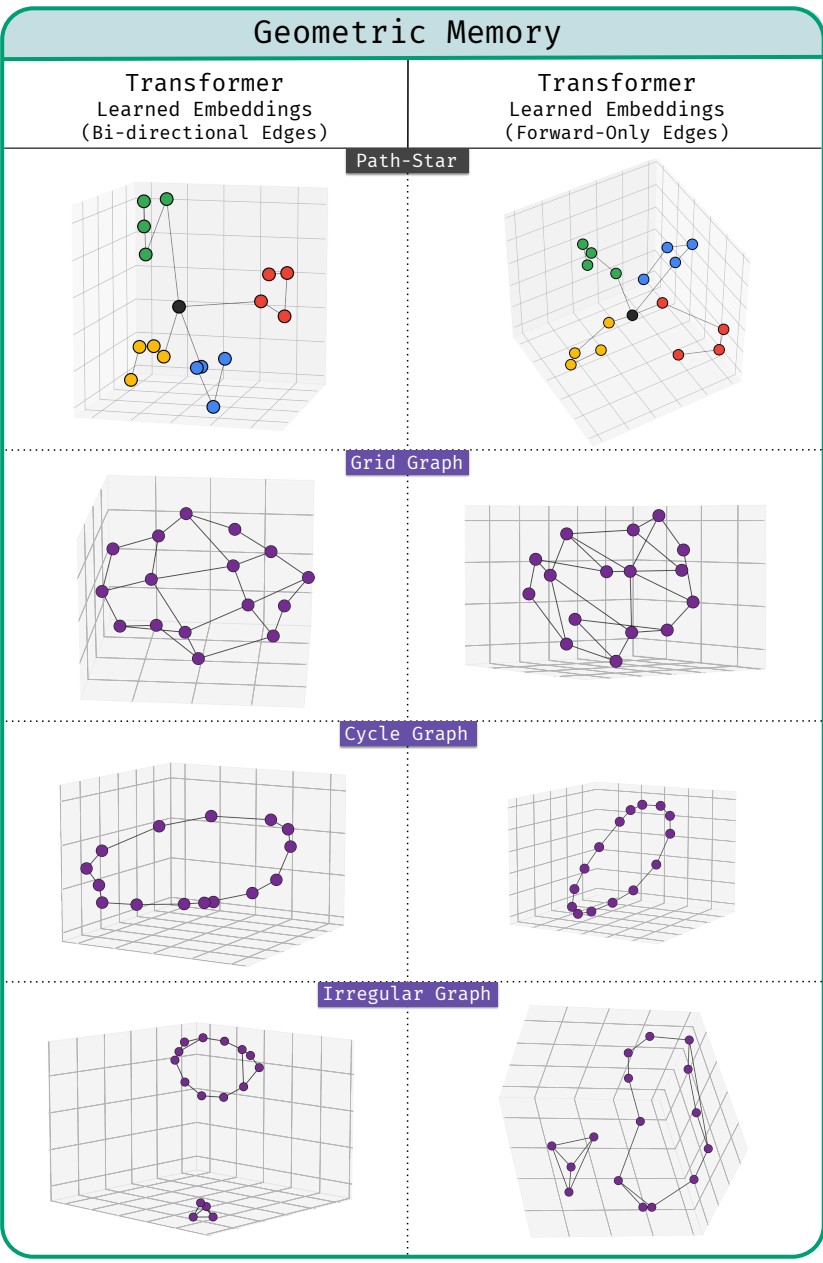

Figure 21: **Embeddings of a Transformer with bi-directional vs. uni-direction edge memorization.** With our smaller graphs, we find a geometry arise regardless of whether the model is made to memorize both or only one direction of each edge. However, the geometry is weaker (e.g., for the grid graph) under uni-directional memorization.

In the same in-weights path-finding task of §2, we find that it is helpful to insert pause tokens (Burtsev et al., 2020; Goyal et al., 2024) to achieve quicker accuracy gains during training. Pause tokens are added by appending dummy tokens to the prefix of the path-finding task both during training and inference.

Fig. 22 shows that adding a short sequence of pause tokens after the prompt reliably boosts exact-match accuracy across graphs, for a given amount of training time. Increasing the number of pause tokens increases speed of convergence.

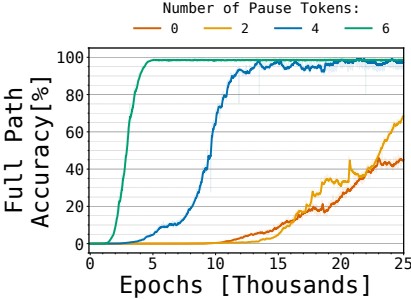

Figure 22: **Pause tokens boost convergence speed of in-weights path-star path-finding task of §2.**

### F.3   (NOT) INTERLEAVING EDGE-MEMORIZATION

In all our experiments, we have interleaved edge-memorization examples with path-finding examples. An alternative training method would be a two-phased approach, where we first enforce edge-memorization, and then follow up by finetuning on the path-finding task. We found this to be less stable, e.g., the model achieves a peak accuracy momentarily, only to deteriorate dramatically right after. This is a manifestation of the well-known effect that finetuning has on parametric memory (Li et al., 2024; Luo et al., 2025). Since this is a confounding effect, we do not choose this regime for our experiments.

However, we confirm that even in this regime, our models do achieve a high peak accuracy (see Fig. 23). The fact that the composition task is learnable in this regime implies that the edge-pretrained model must have come with an adequate global geometry despite being trained only on local supervision (Observation 3b).

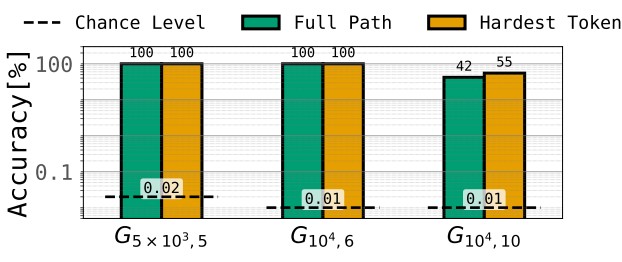

Figure 23: **Locally supervised model succeeds at path-finding task.** We report the *peak* accuracy under finetuning an edge-memorizing model on our path-finding task of §2. We emphasize that this accuracy value is only reached momentarily, and typically deteriorates quickly during further finetuning. Nevertheless, this suggests that local supervision alone was adequate to synthesize a global geometry (Observation 3b). Learning rates of the two phases are given in §D.3.

## F.4 LEARNING ORDER OF TOKENS

For clarity, in Fig. 24, we provide a side-by-side contrast between the order in which tokens are learned in the in-weights setup (with next-token prediction) and the in-context setup (with multi-token prediction).

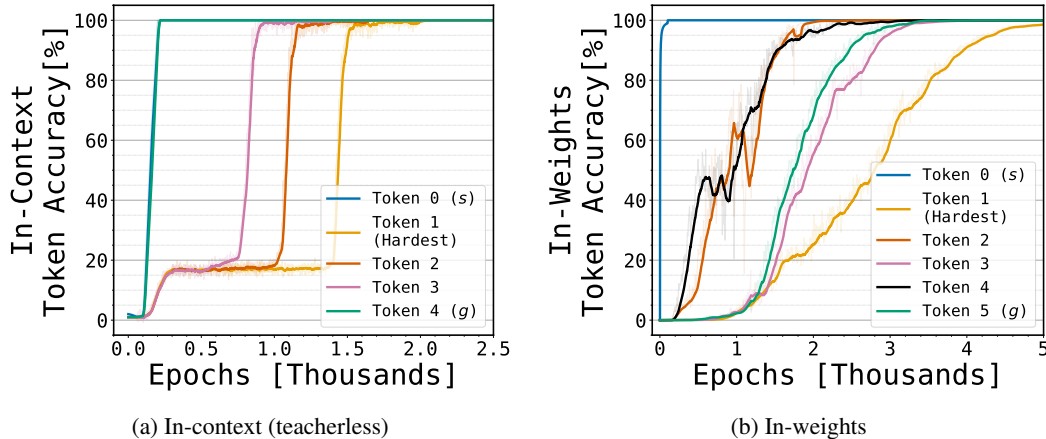

(a) In-context (teacherless)  (b) In-weights

Figure 24: **Learning dynamics per token.** **(a)** In the in-context setting of $\mathcal{G}_{2,5}$ (trained with a multi-token, teacherless objective since standard next-token prediction fails), later tokens are learned first indicating strong reliance on future-token signals. **(b)** In the in-weights setting of $\mathcal{G}_{10^4,6}$ with next-token prediction, token accuracies rise largely in tandem (or in a somewhat confusing order); the first token is not selectively driven by future targets.

## G PROOFS ABOUT REPRESENTATIONAL COMPLEXITY

### G.1 EMPIRICAL FAILURE OF COMPOSITION LEARNING UNDER ASSOCIATIVE MEMORY

As discussed in §2.2, it is well-known that certain compositional tasks are hard to learn empirically; theoretically, this has been proven in a certain sense. However, these prior discussions involve composing information available in the context, rather than in the weights. To test this intuition in our in-weights composition task, we design an experiment where we freeze the embeddings of a Transformer and train it on our path-star task. If the model succeeded in this task, it may mean one of two things: either the model develops a geometric memory in a subsequent layer, or the model does in fact efficiently learn how to compose associative matrix operations—going against our intuition derived from prior limits on composition learning. However, we find that even after $50,000$ training epochs (using the same `GPT-mid` architecture described in Table 2 and §D.2.1, except with frozen token embeddings; and the optimization hyperparameter grid search reported in §D.3), the model fails to learn the in-weights path-star task. We believe, this is preliminary evidence that associative memory indeed struggles to learn compositional in-weights tasks. A fleshed-out proof of this negative result is left for future work.

### G.2 SUCCINCTNESS DOES NOT BREAK THE TIE: PROOF OF PROPOSITION 1

In datasets where redundancies exist, the complexity of a lookup table scales quickly with the training set size (say $n$), whereas the more succinct solution does not (or at worst, grows polynomially slower). For example, if the data is linearly separable in some constant dimensionality, the linear classifier can be described in a constant number of bits, whereas a lookup table (such as a nearest neighbor model) would require $n$ bits. However, this wide disparity in complexity does not necessarily surface in our setting, which is a *memorization* task without redundancies. Concretely, at least in terms of bits and $\ell_2$ norms, there are simple graphs where both the geometric and associative views are equally complex—there is no factor of $n$, which in our case should be the edge count or the vertex count of our graph. We informally prove this below.

We first roughly derive the bit and norm complexity for a general graph, showing how an associative memory scales with the edge count, whereas a geometric memory scales with the vertex count. Then, we argue how this resolves to similar values for graphs like the path-star or a cycle, where the edge count and the vertex count are the same (almost).

**Notation.** We let $|V|$ be the number of entities and $|E|$ the number of associations.

**Proposition 2.** *(**Bit complexity**) Storing a graph $\mathcal{G} = (V, E)$*

- *with associative memory requires $|E| \log |V|$ many bits (with a multiplicative factor of $2$ if both direction of the edges must be stored).*

- *with geometric memory requires $|V| m \log \Delta$ many bits where $m$ is the embedding dimensionality, and $\Delta$ is the number of cells along each dimension required to avoid collision. This is doubled if (un)embedding weights are not tied.*

*Proof.* In the local associative view, given as input any vertex $u$, we must be able to lookup the vertex IDs of its neighbors. Thus, at the position of this vertex, we need a total of $d(u) \log |V|$ many bits (where $d(u)$ is the degree, and $\log |V|$ is the bit length of each ID). Summing this over all vertices gives us $|E| \log |V|$ many bits (since the sum of all degrees must equal the edge count). Note that an extra factor of $2$ appears if both the direction of the edges must be stored.

In the geometric embedding, each vertex is stored as a vector in $m$ dimensions. Each dimension must store one of $\Delta$ values, which requires $\log \Delta$ bits. Summing this up across all dimensions and vertices gives us the result. This is doubled if the unembedding matrix is not weight-tied.

$\square$

**Proposition 3.** *($\ell_2$ **norm complexity**) Given a graph $\mathcal{G} = (V, E)$, and without loss of generality, given the margin constraint that if $u$ is a neighbor of $v$, then $f(u)[v] - \arg\max_{w \notin nbr(u)} f(u)[w] \geq 1$ ($f(u)[v]$ denotes the logit of predicting $v$ given $u$; and $w$ is a non-neighbor), then*

1. *associative memory requires $\ell_2$ norm of at most $\sqrt{|E|}$ with an extra factor of $\sqrt{2}$ if both directions must be stored.*

2. *geometric memory requires an $\ell_2$ norm of at least $\sqrt{|V|}$ with an extra factor of $\sqrt{2}$ if (un)embedding weights are not tied.*

*Proof.* Recall that associative memory takes the form $f(u)[v] = \mathbf{\Phi}(v)^T \mathbf{W}_{\mathrm{assoc}} \mathbf{\Phi}(u)$. Without loss of generality, if we assume that the embeddings are one-hot vectors in $\mathbb{R}^{|V|}$, we can set $\mathbf{W}_{\mathrm{assoc}}$ to be the adjacency matrix to satisfy our margin constraint. The $\ell_2$ norm (of the free parameters in $\mathbf{W}_{\mathrm{assoc}}$) is $\sqrt{|E|}$.

In the geometric view, recall that $f(u)[v] = \mathbf{\Phi}_{\mathrm{geom}}(v)^T \mathbf{\Phi}_{\mathrm{geom}}(u)$. To have $\mathbf{\Phi}_{\mathrm{geom}}(v)^T \mathbf{\Phi}_{\mathrm{geom}}(u) > 1$, we need $\|\mathbf{\Phi}_{\mathrm{geom}}(v)\|^2 + \|\mathbf{\Phi}_{\mathrm{geom}}(u)\|^2 > 2$. Thus, roughly, all embedding norms must be at least 1, implying that $\sum_u \|\mathbf{\Phi}_{\mathrm{geom}}(u)\|^2 \geq |V|$. $\qquad\square$

**Proof of Main Proposition 1**

*Proof.* Our proof follows from the fact that for graphs like the path-star graph and the cycle graph, the edge count and vertex count are nearly equal. Thus, both notions of complexity—the associative scaling with the edge count and the geometric with vertex count—can be shown to reduce to similar values here from the above propositions.

This is straightforward to see for $\ell_2$ norm based complexity based on Proposition 3. For the bit complexity estimates, from Proposition 2, we know that associative memory costs $|V| \log |E|$ many bits; for the geometry, we need to pin down the values of the embedding dimension $m$ and the cell count $\Delta$.

The path-star graph can be embedded such that each path is stored along a unique dimension (thus totally $m = d$ dimensions), and each dimension can be gridded into $\ell$ many cells. This requires a bit complexity of $|V| d \log \ell$. For a more ambitious geometry, we could be further squeeze this into $\log d$ dimensions while still keeping the paths well-separated (by the Johnson–Lindenstrauss lemma), resulting in $|V| \log d \log \ell$ bits. This is still greater than the cost of associative memory which is approximately $|V|(\log d\ell) = |V|(\log d + \log \ell)$.

A similar argument works for a cycle graph. Here, we can embed in $m = 2$ dimensions, with a cell count of $|V|/2$, thus totaling $2|V| \log(|V|/2)$ bits for geometric memory, again greater than $|V| \log |E|$ when $|V| = |E|$. $\qquad\square$

G.3  NODE2VEC CAN REPRESENT A FORM OF ASSOCIATIVE MEMORY

In the standard associative memory view that we have discussed, associations are represented through the function $\mathbf{\Phi}(v)^T \mathbf{W}_{\mathrm{assoc}} \mathbf{\Phi}(u)$. However, dual encoder models like `Node2Vec` can only represent functions of the form $\mathbf{\Phi}(v)^T \mathbf{\Phi}(u)$. While this precludes the form of associative memory we care about, it still allows a contrived form of associative memory when there is a sufficiently large embedding dimensionality. Below we show that given a set of edges $E$, and a dimensionality of $|E|$, we can construct embeddings such that the dot product of two adjacent vertices are high, but any non-adjacent vertices have zero dot product.[2] In this sense, only local information is captured, whereas no global geometry is.

**Proposition 4.** *Dual encoder models like `Node2Vec` can represent memory associatively with an embedding dimensionality of $|E|$ where $E$ is the set of pairwise associations.*

*Proof.* Assume that each node is embedded in an $|E|$-dimensional space, where the $i$th dimension corresponds to the $i$th edge. Then, for edge $(u, v)$ we assume that the embedding of $u$ and $v$ both are set to 1 along the dimension corresponding to $(u, v)$. Notationally, if the edges are indexed as $1, 2, \ldots$, then $\mathbf{\Phi}(u)_i = \mathbf{1}[u \in e_i]$, where $\mathbf{1}$ is the indicator function. Then, we have that $\mathbf{\Phi}(u) \cdot \mathbf{\Phi}(v) = \mathbf{1}[(u, v) \in E]$. Thus, the dot products capture only local information. Any two non-adjacent vertices have a zero dot-product. No worthwhile geometry exists. $\qquad\square$

---

[2]We suspect this should also be a lower bound i.e., such a large dimensionality must be needed to represent associatively in `Node2Vec`.

## H DETAILED ANALYSIS OF SPECTRAL BIAS IN NODE2VEC

Let $G$ be a graph of $n$ nodes $\{1, 2, \ldots, n\}$. Let $\mathbf{A} \in \mathbb{R}^{n \times n}$ be the adjacency matrix, $\mathbf{D} \in \mathbb{R}^{n \times n}$ the diagonal degree matrix, and let the embedding of the nodes be denoted by $\mathbf{V} \in \mathbb{R}^{n \times m}$, where $m$ is the embedding dimensionality. Let $\mathbf{L} = (\mathbf{I} - \mathbf{D}^{-1}\mathbf{A}) + (\mathbf{I} - \mathbf{D}^{-1}\mathbf{A})^T$ denote the *asymmetrically normalized random walk graph Laplacian*. The second topmost eigenvectors of $-\mathbf{L}$ are called the Fiedler vectors; we refer to them and the next few eigenvectors as Fiedler-like eigenvectors. The topmost eigenvector of $-\mathbf{L}$ is a degenerate eigenvector of (approximately) all 1s.

**Node2Vec setup.** We consider the simplest Node2Vec model, where the embeddings are directly parameterized by $\mathbf{V}$. We consider a 1-hop objective (where the neighborhood is defined by the immediate neighbors rather than by more distant ones discovered by a random walk). Note that our objective uses the full softmax loss:

$$\mathcal{J}_{\texttt{Node2Vec}}(\mathbf{V}) = \max_{\mathbf{V}} \sum_i \frac{1}{|\texttt{nbr}(\cdot)|} \sum_{j \in \texttt{nbr}(i)} \log \underbrace{\frac{\exp(\mathbf{v}_i^T \mathbf{v}_j)}{\sum_k \exp(\mathbf{v}_i^T \mathbf{v}_k)}}_{p(i,j)}, \tag{1}$$

where $\texttt{nbr}(\cdot)$ denotes the neighboring vertices in graph $\mathcal{G}$. The above (degree-normalized) objective resembles optimizing over a sequence dataset where we sample the first vertex uniformly, and the second vertex uniformly from its neighborhood.

Let $\mathbf{P} \in \mathbb{R}^{n \times n}$ be the matrix of probabilities $p(i,j)$, where:

$$\mathbf{P} = \texttt{row\_softmax}(\mathbf{V}\mathbf{V}^T). \tag{2}$$

The dynamics of the Node2Vec algorithm can be expressed as below:

**Lemma 5.** *The update on the representations under gradient maximization of the Node2Vec objective in Eq 1 can be written as:*

$$\Delta\mathbf{V}(t) = \eta\mathbf{C}(t)\mathbf{V}(t) \text{ where, } \mathbf{C}(t) = \underbrace{(\mathbf{D}^{-1}\mathbf{A} - \mathbf{P}(t)) + (\mathbf{D}^{-1}\mathbf{A} - \mathbf{P}(t))^T}_{co\text{-}efficient\ matrix} \tag{3}$$

We prove this in §H.5.

### H.1 CHALLENGES OF ANALYZING THE DYNAMICS

Unlike previously-studied dynamics which simplify nicely, this system may behave in one of many ways. For one, it may simply diverge, but if we are a bit lucky, it may at least converge in direction (like in logistic regression (Soudry et al., 2018)); but then, this direction then may be degenerate—an all-one representation could potentially be a stable direction—and perhaps nice directions are visible only if we analyze with early-stopping. One way to get a handle of this would have been to show that in the limit, we have $\mathbf{C}(t) \to 0$; solving this could then spell out the (limit) probability matrix $\mathbf{P}$, if not the inner products $\mathbf{V}\mathbf{V}^T$ themselves. However, we find that $\mathbf{C}$ cannot be zero as that would require the self-probability term $p(i,i) = 0$, which is infeasible. This closes all obvious analytical routes to understanding this system, so we turn to an empirical study.

### H.2 EMPIRICAL INTUITION OF THE DYNAMICS

Empirically, we find that the model tends towards a gradient-zero state by working its way toward satisfying a two-fold constraint in Observation 7. First, the column space of $\mathbf{V}(t)$ converges to the top eigenvectors of $\mathbf{C}(0)$—which is approximately the negative of the Laplacian $\mathbf{L}$. Concurrently, $\mathbf{C}(t)$ itself converges such that its null space matches these eigenvectors. Together then, the update $\Delta\mathbf{V}(t)$ in Lemma 5 must become zero.

**Observation 7.** *We find that $\Delta\mathbf{V}(t) \to 0$ through the following concurrent behaviors:*

- *The null space of $\mathbf{C}(t)$ spans the top eigenvectors of $-\mathbf{L}$.*

- *The column space of $\mathbf{V}(t)$ converges to the top eigenvectors of $-\mathbf{L}$.*

Crucially, we find that this can happen (a) even without a constraint on the dimensionality $m$ and (b) this requires *no* early-stopping (see Remark 2 for a more nuanced discussion of this).

We lay out our empirical intuition below, deferring a more mathematical description of the same to the following section. First, we postulate a key invariant during training: the eigenvectors of the co-efficient matrix $\mathbf{C}(t)$, the probability matrix $\mathbf{P}(t)$ and the embeddings $\mathbf{V}(t)$ all remain (inexplicably) stable during training. In particular, since the system begins with $\mathbf{P}(0) \approx \mathbf{I}$, and so $\mathbf{C}(0) \approx -\mathbf{L}$ all these eigenvectors are then fixed as the eigenvectors of the normalized random walk graph Laplacian .

Next, we find that the eigenvalues of the co-efficient matrix $\mathbf{C}(t)$ begin *negative*, gradually approaching zero. The top eigenvectors reach zero first, achieving the second condition in Observation 7. That the values begin negative follows from the fact that the co-efficient matrix begins as the negative graph Laplacian. That these values approach zero follows from the fact that embedding vectors become less orthogonal over time; this in turn reduces the eigenvalues of $\mathbf{P}(t)$, which increases the eigenvalues of $\mathbf{C}(t)$.

Next, due to the negative eigenvalues of $\mathbf{C}(t)$, the embeddings $\mathbf{V}$ along the lowermost eigendirections quickly diminish, achieving our first condition. Note that this means we do not want early-stopping; unlike in the quadratic loss formulation of Karkada et al. (2025), it is longer training that filters out the lower eigenvectors. (Although, the existence of a degenerate eigenvector complicates this; see Remark 2). This achieves the first condition in Observation 7.

Observe that this argument does not require any upper bound on the size of the embedding space. It is unclear if a more succinct, margin-maximizing or norm-minimizing view of these dynamics is expressible.

**Remark 2.** *(**The degenerate vector and early-stopping**) When the graph Laplacian is symmetrically normalized (e.g., $\mathbf{D}^{-1/2}\mathbf{A}\mathbf{D}^{-1/2}-\mathbf{I}$), the top-most eigenvector of the graph Laplacian is a degenerate vector that assigns a constant value to all nodes, and provably corresponds to a zero eigenvalue. However, in our setting, this eigenvalue is slightly above zero, likely due to the asymmetric nature of our Laplacian. Therefore, as we train for longer, the model would become degenerate thus requiring early-stopping. However, this is a conceptually different reason to early-stop than the one in Karkada et al. (2025). Here we may need to early-stop to prevent collapse to the top eigenvector, whereas in Karkada et al. (2025), it is to prevent expansion to bottom eigenvectors.*

## H.3 MATHEMATICAL DESCRIPTION

Below, we provide a more mathematical description of the above summary by dividing it up into various propositions. Our proofs for these propositions are highly informal. However, our propositions hold in practice without our simplifying assumptions (at least in the graphs we study). We leave it for future work to deliver a rigorous proof and a more clearly characterized theorem statement.

First, we note that the co-efficient matrix approximately begins as the negative graph Laplacian for an appropriately large initialization. (Without this assumption, we may still make a connection to a graph Laplacian-*like* object).

**Assumption 1.** We assume a sufficiently large magnitude or embedding dimensionality of random initialization such that the initial embeddings are nearly orthogonal as $\mathbf{V}(0)\mathbf{V}(0)^T \approx c\mathbf{I}$.

**Fact 1.** *Under Assumption 1,*

$$\mathbf{C}(0) \approx -\mathbf{L} = (\mathbf{D}^{-1}\mathbf{A} + (\mathbf{D}^{-1}\mathbf{A})^T - 2\mathbf{I}). \tag{4}$$

*Proof.* At time $t = 0$, the embeddings $\mathbf{V}(0)$ are all random, and hence nearly orthogonal to each other i.e., $\mathbf{V}(0)\mathbf{V}(0)^T \approx c\mathbf{I}$, where $c$ is some constant that depends upon the magnitude of the random initialization. Since $\mathbf{P} = \texttt{row\_softmax}(\mathbf{V}\mathbf{V}(\texttt{t}))$, for a sufficiently large $c$, $\mathbf{P}(0) \approx \mathbf{I}$, proving our claim. $\qquad\square$

Next, we make the empirical observation that the eigenvectors of $\mathbf{P} + \mathbf{P}^T$ match the eigenvectors of the embedding inner products $\mathbf{V}\mathbf{V}^T$. Note that $\mathbf{P}$ is related to the inner product via a non-linear row softmax operation, rendering a proof of this observation highly non-trivial. We assume this observation (without even an intuitive proof) for the rest of our discussion.

**Observation 8.** *(Eigenvectors remain unchanged under a row-softmax transform) The eigenvectors of $\mathbf{P}(t) + \mathbf{P}(t)^T$ at any time $t$, are also approximately the eigenvectors of the embeddings $\mathbf{V}(t)\mathbf{V}(t)^T$, appearing in the same order.*

From the above observation, we can conclude that the eigenvectors of the system match the Laplacian throughout training. This follows by how the updates reduce to muplications between matrices sharing the same eigenspaces.

**Proposition 6.** *(Time-invariant eigenvectors match that of the Laplacian) With Assumption 1 and by assuming Observation 8 as a given, we have that for all $t$, the quantities $\mathbf{C}(t), \mathbf{P}(t) + \mathbf{P}(t)^T, \mathbf{V}(t)\mathbf{V}(t)^T$ have the same eigenvectors as that of the negative Laplacian $-\mathbf{L}$.*

*Proof.* At any time $t$, we can write the embedding vectors as

$$\mathbf{V}(T) = \prod_{t=0}^{T-1}(1 + \eta\mathbf{C}(t))\mathbf{V}(0), \tag{5}$$

and so the inner product as

$$\mathbf{V}(T)\mathbf{V}(T)^T = \prod_{t=0}^{T-1}(1 + \eta\mathbf{C}(t))\underbrace{\mathbf{V}(0)\mathbf{V}(0)^T}_{\approx c\mathbf{I} \text{ by Assumption 1}}\prod_{t=0}^{T-1}(1 + \eta\mathbf{C}(t))^T \tag{6}$$

$$\approx c\prod_{t=0}^{T-1}(1 + \eta\mathbf{C}(t))(1 + \eta\mathbf{C}(t))^T. \tag{7}$$

From here, we inductively prove our claim. At $t = 0$, it is indeed the case that $\mathbf{C}(t), \mathbf{P}(t), \mathbf{V}(t)\mathbf{V}(t)^T$ all have the same eigenvectors as $\mathbf{L}$ either by Fact 1 for $\mathbf{C}(0)$, or trivially since $\mathbf{P}(t)$ and $\mathbf{V}(t)$ are orthogonal matrices. We assume this is true for all $t$ until $T - 1$. Then, by the above equation, it is also true that the inner product $\mathbf{V}\mathbf{V}^T$ shares these eigenvectors. By invoking Observation 8, we can say that the same is true of the probability matrix $\mathbf{P} + \mathbf{P}^T$. Subsequently, this is true of $\mathbf{C}(t)$, which equals $\mathbf{D}^{-1}\mathbf{A} + (\mathbf{D}^{-1}\mathbf{A})^T + (\mathbf{P} + \mathbf{P}^T)$. (Note that the first term here has the same eigenvectors as $-\mathbf{L}$ as it is off only by the identity matrix.) This proves our inductive assumption. $\qquad\square$

Next, we begin to bound the eigenvalues of the system. For the sake of our informal proofs we make some simplifying assumptions that make our matrices approximately symmetric; however, we do not need these assumptions in practice.

**Assumption 2.** For theoretical convenience, we assume that:

- $\mathbf{P} \approx \mathbf{P}^T$.

- the embeddings (i.e., the rows of $\mathbf{V}$) are of equal $\ell_2$ norms.

- the degrees of all nodes are roughly equal.

Now, we can observe a bound on the eigenvalues of the probability matrix.

**Proposition 7.** *(Eigenvalues of the probability matrix) Under Assumption 2, the eigenvalues of $\mathbf{P}(t) + \mathbf{P}(t)^T$ are such that:*

1. *their sum is upper bounded by $2n$ (where $n$ is the number of nodes).*

2. *they are each approximately bounded in $[0, 2]$*

*Proof.* For the first result, recall the fact the sum of eigenvalues is the trace of the matrix. Since each diagonal term is at most 2 (it is $2p(i, i)$), the trace is atmost $2n$. This requires no special assumptions.

For the bounds on each eigenvalue, we can rely on the Gershgorin Circle theorem, which states that the eigenvalues lie in the union of discs centered at the diagonals $p(i, i)$, each with radius equal to the sum of the absolute off-diagonal terms, $\sum_{j \neq i} p(i, j)$. The upper bound is then equal to the sum of the

rows. For $\mathbf{P}(t)$, this sum is equal to 1 due to the row-softmax operation. Assuming $\mathbf{P}^T \approx \mathbf{P}$—which is approximately true in practice, especially if the node degrees are uniform (but not always)—, we can conclude that the upper bound is approximately 2.

For the lower bound, if we have that the self-probabilities $p(i,i)$ are the largest in any row, then the lower bound $p(i,i) - \sum_{j \neq i} p(i,j)$ is at least zero. This is indeed the case if the embeddings of all nodes are of approximately equal norms, in which case the inner product $\mathbf{V}\mathbf{V}^T$ is highest along the diagonal. $\qquad\square$

**Proposition 8.** *The eigenvalues of $\mathbf{D}^{-1}\mathbf{A} + (\mathbf{D}^{-1}\mathbf{A})^T$ approximately lie in $[-2, 2]$ assuming that the nodes have approximately uniform degree as in Assumption 2.*

*Proof.* The diagonal of $\mathbf{D}^{-1}\mathbf{A}$ is 0 (assuming no self-loops in the graph), while the off-diagonal values are all positive and sum up to 1 in each row. When the node degrees are approximately uniform, $\mathbf{D}^{-1}\mathbf{A} \approx (\mathbf{D}^{-1}\mathbf{A})^T$. From the Gergshgorin circle theorem, the eigenvalues lie in the union of discs centered at the diagonal (from the above, 0) with radii equal to the sum of the absolute off-diagonal terms (from the above, 2), thus proving our claim. $\qquad\square$

Now, we can establish the conditions in Observation 7, namely, the convergence of the null space of the co-efficient matrix from Observation 1c, and then the convergence of the embedding vectors.

**Proposition 9.** *Under Assumption 2 and Assumption 1, at any time instant $t$, the eigenvalues of $\mathbf{C}(t)$ are all strictly negative (except for the topmost eigenvalue, which is of a degenerate all-1 eigenvector, and is approximately zero), and this is so until when the top eigenvectors of the Laplacian converge into the null space of $\mathbf{C}(t)$ (i.e., their eigenvalues become zero).*

*Intuition.* Recall that $\mathbf{C}(t) = \mathbf{D}^{-1}\mathbf{A} + (\mathbf{D}^{-1}\mathbf{A})^T - (\mathbf{P} + \mathbf{P}^T)$. The eigenvalues of the first term lie approximately in $[-2, 2]$ from Proposition 8, while that of the probability term lie in $[0, 2]$, from Proposition 7. Note that eigenvalues of the probability matrix all begin uniformly at 2 in the beginning (as $\mathbf{P}(0) = \mathbf{I}$, by Fact 1 under Assumption 1), as a result of which the initial eigenvalues of $\mathbf{C}(t)$ start at or below 0.

While the embeddings are initialized orthogonally under Assumption 1, they become less orthogonal during training, leading to a gradual decrease of the diagonal self-probability terms in $\mathbf{P} + \mathbf{P}^T$. Intuitively, this also means that the eigenvalues of $\mathbf{P} + \mathbf{P}^T$ must themselves all decrease from the initial value of 2 (based on Proposition 7). In turn, the eigenvalues of $\mathbf{C}(t)$, which begin negative must gradually inch toward zero. The topmost eigenvalues—which are closest to zero—are the first to reach zero.[3] $\qquad\square$

**Proposition 10.** *(Embeddings converge to top eigenvectors) Assuming Observation 8 as a given, for a sufficiently small learning rate $\eta$, with increasing timestep $t$, the column space of $\mathbf{V}(t)$ converges to the top eigenvectors of the negative graph Laplacian $-\mathbf{L}$, independent of the embedding dimensionality.*

*Proof.* We can examine the dynamics of each embedding dimension separately.[4] For the embedding dimension $j = 1, 2, \ldots, m$, let $\mathbf{r}_j \in \mathbb{R}^n$ denote the $j$th column of the embedding matrix $\mathbf{V}$. The dynamics of this column (we drop the index $j$ for the moment) at any timestep $t$ can be isolated as:

$$\mathbf{r}(t) = \prod_{t=0}^{T}(1 + \eta\mathbf{C}(t))\mathbf{r}(0). \tag{8}$$

Given that $\mathbf{C}(t)$ have time-invariant eigenvectors (by Proposition 6), this can be further simplified as

$$\mathbf{r}(t) = \mathbf{E}\left(\prod_{t=0}^{T}(1 + \eta\mathbf{\Lambda}(t))\right)\mathbf{E}^T\mathbf{r}(0). \tag{9}$$

---

[3]Note that this is not straightforward to show. It is possible that the even if the initial eigenvalues are very close to zero, they approach 0 slower than farther off values.

[4]This is possible only in a dual-encoder, `Node2Vec` style architecture. In a Transformer for example, there are cross-dimensional interactions, due to the associative weight matrix $\mathbf{W}_{\text{assoc}}$ that interfaces between the embedding and unembedding layers.

Given that the eigenvalues in $\mathbf{\Lambda}$ are all less than or equal to zero (by Proposition 9), the term $(1 + \eta\mathbf{\Lambda}(t))$ must consist of a diagonal of values in $[0, 1]$ for an appropriately small learning rate. Furthermore, as $T$ becomes large, the values of the top eigenvectors (which have the least eigenvalues, and therefore, the largest value of $1 + \eta\lambda_i(t)$) must come to dominate. Then, as $t$ increases, we can express the embedding dimension as an affine combination of some top $K$ eigenvectors (where the coefficients depend on how the embedding dimension was initialized):

$$\mathbf{r}(t) \approx \sum_{k=1}^{K}\left(\prod_{t}(1 + \eta\lambda_k(t))\mathbf{r}(0) \cdot \mathbf{e}_k\right)\mathbf{e}_k. \tag{10}$$

$\square$

## H.4 EMPIRICAL VALIDATION ACROSS GRAPH TOPOLOGIES

To demonstrate the generality of this spectral bias phenomenon, we validate the convergence dynamics across multiple graph types beyond the path-star graphs analyzed in Fig. 4.

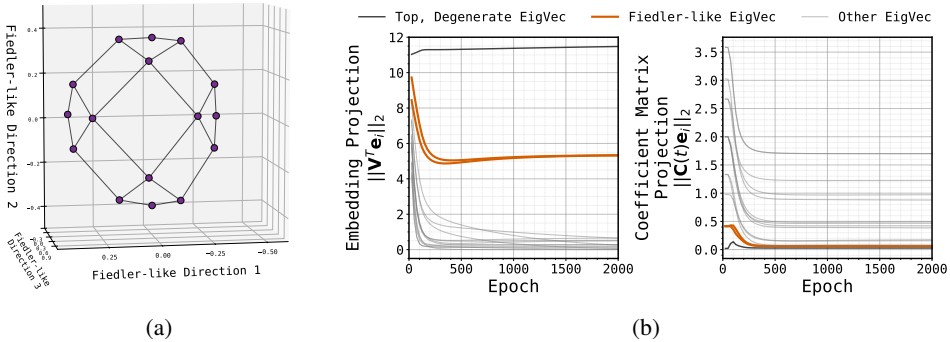

(a)                                      (b)

Figure 25: **Spectral bias emergence in `Node2Vec` for grid graphs. (a)** The Fiedler vectors of a grid graph capture spatial locality and connectivity patterns, closely mirroring the `Node2Vec` embedding shown in Fig. 1 (bottom-right), where a similar geometry emerges. **(b)** Training dynamics show the same two-fold convergence pattern as path-star graphs, with embedding-Fiedler alignment and Fiedler vectors entering the co-efficient matrix null space.

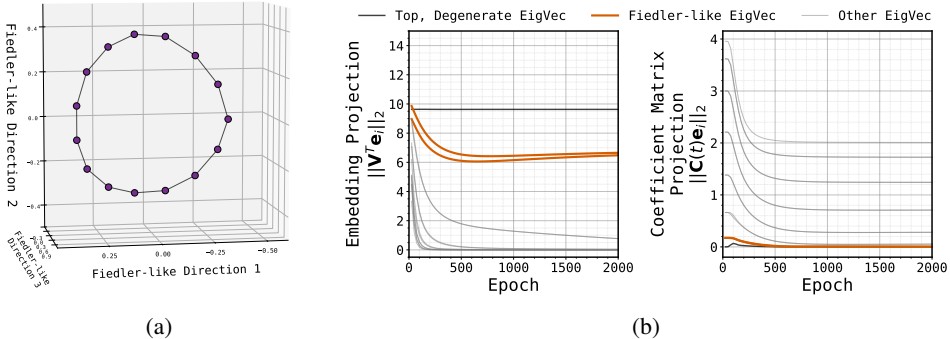

(a)                                      (b)

Figure 26: **Spectral bias emergence in `Node2Vec` for cycle graphs. (a)** The Fiedler vectors of a cycle graph reflect the circular structure with smooth transitions, closely mirroring the `Node2Vec` embedding shown in Fig. 28 (top-right), where a similar geometry emerges. **(b)** Despite the different topology, the same spectral convergence dynamics emerge, confirming that the mechanism is not specific to tree-like structures.

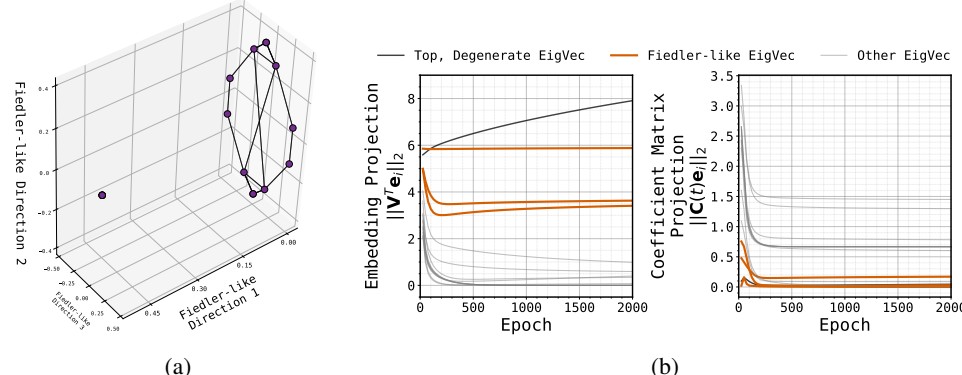

(a)                                        (b)

Figure 27: **Spectral bias emergence in `Node2Vec` for random graphs. (a)** Even in random graphs without clear structural patterns, Fiedler vectors capture the most significant connectivity patterns. **(b)** The spectral convergence dynamics persist, demonstrating robustness across graph types.

Figures 25, 26, and 27 together with Fig. 4 demonstrate that the spectral convergence phenomenon holds across diverse graph topologies. In each case, we observe the same elegant dynamics where embeddings align with the graph's Fiedler-like eigenvectors while these vectors simultaneously move into the null space of the co-efficient matrix, independent of both embedding dimensionality and graph structure.

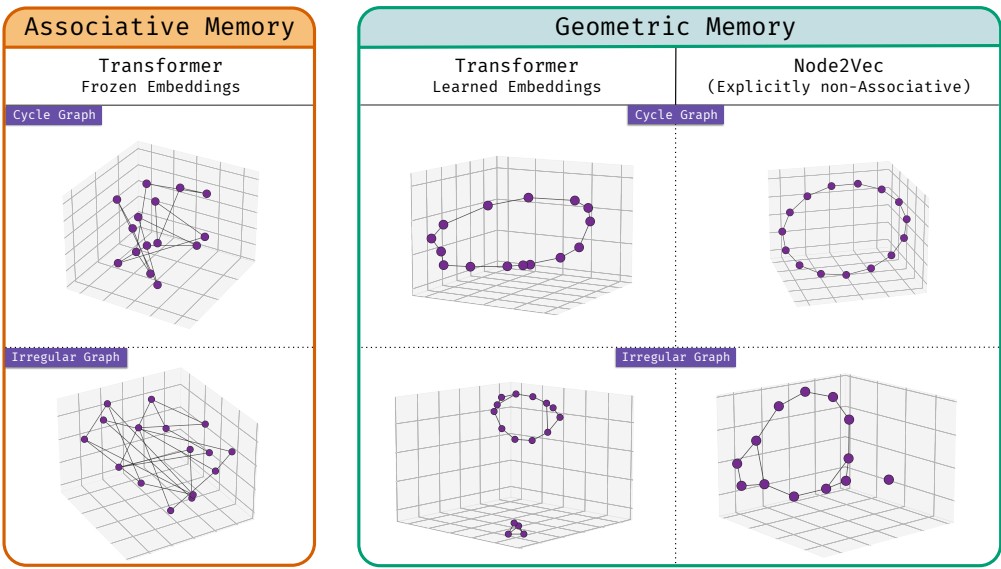

Figure 28: **Quality of geometric memory across graph types.** The contrast between associative memory (left), Transformer geometric memory (middle), and `Node2Vec` geometric memory (right) holds across different graph topologies. `Node2Vec` models, where associative memory is architecturally prohibited, consistently develop cleaner and more structured geometric representations than Transformers. This comparison reveals significant headroom for improving the geometric nature of Transformer memory. Results shown for cycle graphs (top) and random graphs (bottom) complement the path-star and grid-graph analysis from Fig. 1. Details of the Transformer architecture used for this visualization are provided in §D.2.2.

## H.5 DERIVING THE DYNAMICS

We provide proof of Lemma 5 which expresses the dynamical system of our `Node2Vec` objective in Eq 1.

*Proof.* For a pair of nodes with embeddings $\mathbf{u} \in \mathbb{R}^m, \mathbf{v} \in \mathbb{R}^m$, the probability value of the edge $(\mathbf{u}, \mathbf{v})$ can be written as:

$$p(\mathbf{u}, \mathbf{v}) = \frac{\exp(\mathbf{u} \cdot \mathbf{v})}{\sum_{\mathbf{v}'} \exp(\mathbf{u}, \mathbf{v}')}. \tag{11}$$

Let $N(u)$ denote the neighborhood of the node $u$, and $N_u$, its degree. Let $\mathcal{J}_u$ denote the summand in the objective function specific to that node:

$$\mathcal{J}_u(\mathbf{V}) = \frac{1}{|N(u)|} \sum_{\mathbf{v} \in N(u)} \log p(\mathbf{u}, \mathbf{v}) \tag{12}$$

We now compute the derivative of $\mathcal{J}_u$ with respect to itself $\mathbf{u}$:

$$\frac{\partial \mathcal{J}_u(\mathbf{V})}{\partial \mathbf{u}} = \frac{1}{N_{\mathbf{u}}} \sum_{\mathbf{v} \in N(u)} \left( \underbrace{\mathbf{v}}_{\text{numerator}} - \underbrace{\sum_{\mathbf{v}' \neq \mathbf{u}} p(\mathbf{u}, \mathbf{v}')\mathbf{v}' - 2p(\mathbf{u}, \mathbf{u})\mathbf{u}}_{\text{denominator}} \right) \tag{13}$$

$$= \frac{1}{N_{\mathbf{u}}} \sum_{\mathbf{v} \in N(u)} \mathbf{v} - \sum_{\mathbf{v}' \neq \mathbf{u}} p(\mathbf{u}, \mathbf{v}')\mathbf{v}' - 2p(\mathbf{u}, \mathbf{u})\mathbf{u} \tag{14}$$

Next, we compute the derivative of $\mathcal{J}_w$ with respect to $\mathbf{u}$ for nodes $w \neq u$ :

$$\frac{\partial \mathcal{J}_w(\mathbf{V})}{\partial \mathbf{u}} = \frac{1}{N_w} \sum_{\mathbf{v} \in N(w)} \left( \underbrace{\mathbf{1}[u = v]\mathbf{w}}_{\text{numerator}} - \underbrace{p(\mathbf{w}, \mathbf{u})\mathbf{w}}_{\text{denominator}} \right) \tag{15}$$

$$= \frac{1}{N_w} \sum_{\mathbf{v} \in N(w)} \mathbf{1}[u = v]\mathbf{w} - p(\mathbf{w}, \mathbf{u})\mathbf{w} \tag{16}$$

$$\tag{17}$$

By writing the above expressions as a matrix formula, we get the dynamical system claimed in Lemma 5. $\qquad\square$

