# OpenReview forum: "Transformers tend to memorize geometrically; it is unclear why."
_ICLR.cc/2026/Conference — ICLR 2026 Conference Withdrawn Submission_

### Official Review · Reviewer_2EuG · 2025-10-26

**Soundness:** 2
**Presentation:** 3
**Contribution:** 3
**Rating:** 4
**Confidence:** 4

**Summary:**

This paper implies a geometric form of memory in Transformer to encode global structure information rather than only local associations. Experiment results demonstrate promising results

**Strengths:**

1. Paper is well written and easy to follow
2. Experiment is good and result is encouraging

**Weaknesses:**

I have my concern in the IN-WEIGHTS PATH-STAR TASK experiment. In this experiment, there is some information leakage, since training and testing is done in the same graph. In order to claim "IN-WEIGHTS REASONING IS LEARNED", testing should be done in new graph (with new graph topology), or with permutated labels for the node.

**Questions:**

1. see above weakness
2. In section 4, spectral analysis is mainly done in Node2vec, any connection with Graph Laplacian for the Transformer side?

---

> ### Author Response · Authors · 2025-11-22
> **Response**
>
> Dear reviewer,
>
> > in this experiment, there is some information leakage, since training and testing is done in the same graph.  In order to claim "IN-WEIGHTS REASONING IS LEARNED", testing should be done in new graph
>
> **This is incorrect.** Our model only sees a subset of the paths, besides all the edges; we test on finding unseen paths which is a non-trivial search task. Similar and much simpler in-weights path-finding tasks have been studied in the papers we cite (Khona et al. (2024); Ye et al. (2025); Wang et al., 2024a).
>
>
> > spectral analysis is mainly done in Node2vec, any connection with Graph Laplacian for the Transformer side?
>
> Our new experiments imply that the spectral bias is fundamental to any deep sequence model, including neural networks. Thus our Node2Vec analysis isolates the core of this bias. We explain this in common response under `The simplicity of the Node2Vec model is a strength.`.

---

### Official Review · Reviewer_eCS4 · 2025-10-26

**Soundness:** 3
**Presentation:** 1
**Contribution:** 2
**Rating:** 2
**Confidence:** 3

**Summary:**

This paper investigates how Transformers memorize and reason over graph-structured data stored in their weights. The authors present a "path-star" graph task where, contrary to expectations, the model succeeds at finding "in-weights" paths. They posit this success is due to the emergence of "geometric memory" (where embeddings encode global graph structure) rather than "associative memory," or a simple lookup of local facts. The authors argue that the emergence of this geometric structure is not explained by standard pressures (like simplicity bias) and hypothesize that it stems from a "spectral bias," which they explore in the context of a simpler Node2Vec model.

**Strengths:**

1. The paper presents an interesting phenomenon which explores in-context learning and an alternative form of memorization via their experimentation of the path-star task.


2. The paper articulates the difference between "associative memory" (local, pairwise) and "geometric memory" (global, structural), offering a helpful lens for analyzing how models store knowledge.

**Weaknesses:**

1. The main explanation for the phenomenon (spectral bias) is explored only in a simpler Node2Vec model. The paper fails to provide a convincing experimental or theoretical link to show that this same mechanism is responsible for the results observed in the Transformer.

2. The visual evidence presented for the geometric memory in the Transformer is unconvincing. The UMAP plots (e.g., Fig 3b, Fig 10) are messy and do not show the clear, well-separated clustering that the text claims, forcing the argument to rely on heatmap interpretations.

3.  The paper sets up a compelling puzzle (success in Transformers) but then pivots entirely to a different model (Node2Vec) for a solution. This jump makes the overall argument feel disconnected and leaves their central question ultimately unanswered and confusing.

**Questions:**

1. Why should we assume that the spectral bias observed in a 1-layer Node2Vec model is the primary mechanism at play in a complex, multi-layer Transformer? Could the observed geometric memory in Transformers be an emergent property of the attention mechanism?

2. Could the usage of standard sequential positional encoding undermine the claim of geometric memory? The model is learning a node's position in a sequence, not its inherent position in a graph, which seems to reinforce the very "associative" view the paper claims to refute.

Side note:
1. Your Fig. (1) has a weird error. The caption "Node2Vec Associative Memory..." on the left orange figure has overlapping texts.

---

> ### Author Response · Authors · 2025-11-22
> **Response**
>
> > Could the observed geometric memory in Transformers be an emergent property of the attention mechanism?”
>
> Great question! Since our original submission, we have experimented further to ask exactly this. Our findings show that this is not unique to attention mechanisms. In new experiments, we show that even Mamba and Neural networks (without an attention layer) show the same geometries as Transformers and Node2vec. Please see `Fig 12-Fig 15` and also `E.1`.
>
>
> > is explored only in a simpler Node2Vec model, pivots entirely to a different model (Node2Vec) for a solution,
>
> The fact that our analysis is on Node2Vec is in fact a strength since it establishes the fundamental nature of the spectral bias without fancy architectural additions. We discuss this in the common response under `The simplicity of the Node2Vec model is a strength`.
>
>
> > Could the usage of standard sequential positional encoding undermine the claim of geometric memory?
>
> This is a keen observation. In experiments where we train only on local edges (on both directions), there is no helpful position bias e.g.,`Fig 3c` or `Fig 1`. Thus, we have already ruled out this possibility. We will update the paper to include this detail.
>
>
> > “ leaves their central question ultimately unanswered and confusing”
>
> We believe that our contribution is carefully formulating a memorization puzzle in sequence modeling and in considering various possible answers and ruling them out. Please see our common response under `On the lack of a (clear) answer`.

---

> > ### Comment · Reviewer_eCS4 · 2025-11-25
> >
> > # Summary
> > I thank the authors for their response. However, I remained **unconvinced** about their results (on the main message regarding geometry memory vs. associative memory). I am nonetheless thankful that they did the experiment to rule out the effects of attention mechanism, by contrasting on other architectures like MAMBA. I will raise my score from 2 -> 4 as it is fair to do so, however.
> >
> > # Citations
> > I first would like to suggest additional citations relating to associative memory systems:
> >
> > [Hopfield (1982)](https://www.pnas.org/doi/10.1073/pnas.79.8.2554), [Krotov and Hopfield (2016)](https://proceedings.neurips.cc/paper_files/paper/2016/hash/eaae339c4d89fc102edd9dbdb6a28915-Abstract.html),
> > [Ramsauer et al. (2020)](https://arxiv.org/abs/2008.02217),
> > [Hoover et al. (2023)](https://proceedings.neurips.cc/paper_files/paper/2023/file/57a9b97477b67936298489e3c1417b0a-Paper-Conference.pdf), [Smart et al. (2025)](https://arxiv.org/abs/2502.05164)
> >
> > The reasoning behind these suggestions is the following: the people who are working in this field are actively trying to uncover the nature of different types of memorization (via the usage of energy-based modeling), and how it can live at the edge of generalization as well (which I believe you are trying to demonstrate in your paper).
> >
> > # Questions
> >
> > 1. You mention that you train your models only on local bidirectional edges and there exists no helpful position bias. What exactly do you mean by this? Are you saying that due to your experimental setup you would violate it if you use, for example, laplacian positional encoding or other forms of positional encoding on graphs (see this [paper](https://arxiv.org/pdf/2205.12454))?
> >
> > 2. Regarding edge memorization, how does the attention heat map or even the cosine-similarity heat map of embeddings correlate (visually) to that of the structure of the given graph or nodes in a graph (in-context vs non-in-context)?
> >
> > 3. I understand that the work focuses on Transformer but what about GNNs and Graph Transformers. Such models are certainly different but it could be worth while to provide a short discussion as well.

---

> > > ### Author Response · Authors · 2025-11-28
> > > **Response to questions**
> > >
> > > Dear reviewer,
> > >
> > > We are happy to hear your follow-up questions. Please find our answers below:
> > >
> > > > there exists no helpful position bias. What exactly do you mean by this? Are you saying that due to your experimental setup you would violate it if you use, for example, laplacian positional encoding or other forms of positional encoding on graphs
> > >
> > > 1. That's right: adding any special positional graph-based encodings OR providing full path supervision would explicitly feed global information into the model. This is cleanly ruled out in the local edge-memorization experiments. These experiments show that the model synthesizes the geometry without explicit encoding/supervision to do so.
> > >
> > > >  how does the attention heat map or even the cosine-similarity heat map of embeddings correlate (visually) to that of the structure of the given graph or nodes in a graph
> > >
> > > 2. The strong diagonal cell value at cell [i,i] indicates that the leaf node and the first node in path [i] are very closely embedded; the off-diagonal values are all weak, meaning that the leaf node of path i, and first node of path j are far away in embedding space. This captures the key structure in the path-star structure: the disjoint paths. Does this answer your query?
> > > (We do not study any notion of geometry in the original in-context path-star task since that's a fundamentally different task.)
> > >
> > >
> > > > what about GNNs and Graph Transformers. Such models are certainly different but it could be worth while to provide a short discussion as well.
> > >
> > > 3. We have made a mention of graph transformers in Line 1529. These are an altogether different task since the graph is specified as input, not baked in the weights.
> > >
> > > > suggest additional citations relating to associative memory systems...
> > >
> > > Thank you for the suggested citations, and your reasoning for those. **We note that we have already cited all but the last in the appendix (see B.4, what was B.3)**, due to space constraints (we have discussed the more directly related associative memory papers in the main paper, like Bietti et al.,). We will add a mention of Smart et al., too, thank you!

---

### Official Review · Reviewer_9RrG · 2025-10-27

**Soundness:** 3
**Presentation:** 1
**Contribution:** 2
**Rating:** 2
**Confidence:** 3

**Summary:**

The paper proposes an in-weights variant of the path star task, where the goal is to memorize a star graph topology (in existing approaches the graph varies and is not memorized). Similarly to existing approaches, the model is trained on paths  from leafs to the center node and is evaluated on paths from the center to the leaves. Notably, in this setting the model generalizes to the test case (this is not the case in the  original path-star task). The authors investigate reasons for why this happens and argue that transformer memory works geometric.

**Strengths:**

-  **(S1) Novelty,  Contribution:**  The in-memory star-path task is novel and the observation that the model generalizes well is interesting.
- **(S2) Significance:** Understanding generalization capability of transformers is an important problem.
- **(S3) Experiments:** Empirically testing the different hypothesis on transformer memory / generalization is interesting. This paper contains several good observations. In particular: the model implicitly learns distances between nodes without being explicitly trained  for it.

**Weaknesses:**

While I think that the empirical work in this paper is solid, there are significant problems in the writing and presentations that hold this paper back:

- **(W1) Problem Presentation:** The problem central to this paper (Path-Star task) is not defined properly in the paper. While a good explanation exists, it is buried on page 25 in the appendix. This makes it difficult to understand the paper. In contrast, the paper devotes almost half a page to Figure 1 which is difficult to understand as it requires 10 lines of caption. As a side-note: What does the color in  Figure 1  encode?

- **(W2) Writing:** The writing often makes it difficult to understand the paper. Consider this:
>Early on during training, the model fits the later tokens in the target —
concretely nodes $v_2, \ldots , v_{\text{goal}}$ — as the unique child of the previous token provided in the input
during next-token training. This is known as a Clever Hans cheat: the model uses a local rule that
relies on witnessing part of the ground truth prefix $(p, r<i)$ to predict the next token $r_i$ — as against
only using the prefix p to predict the answer tokens

From this explanation, it is not clear what the "Clever Hans" cheat actually is. In contrast, the paper that introduces this problem [The Pitfalls of Next-Token Prediction](https://arxiv.org/pdf/2403.06963) makes the cheat quite clear. Furthermore, sentences are often convoluted making it  difficult to follow:
> Next, we cast doubt on statistical pressures by scrutinizing the applicability of simplicity bias


- **(W3) Significance:** The paper is not clear on what its contributions are. It is mainly a flood of hypotheses, observations and related works. From the abstract:
>  Our insight is that global geometry arises from a spectral bias that—in contrast to prevailing intuition—does not require low dimensionality of the embeddings. Our study raises open questions concerning
implicit reasoning and the bias of gradient-based memorization, while offering a simple example for analysis. Our findings also call for revisiting theoretical abstractions of parametric memory in Transformers.

What does this mean? While the structure "Hypothesis" -> "Observation" is interesting it makes it very difficult to follow. For example, Observations often weaken Hypothesis requiring extra explanation after the observation.

-**(Minor Weakness) Node2Vec:** The section about Node2Vec seems a bit orthogonal to the rest of the paper.

**To sum up,** the combination of a weak presentation, unclear writing and the paper not being clear about its own contributions makes it difficult for me to determine what the paper really contributes. I recommend that the authors significantly rewrite their text with a focus on clarity. As I do not believe that this can be done during rebuttal, I vote for reject. Furthermore, it seems that related work and connections to literature are present in many positions in the paper, making it difficult to  judge what is novel and what is not.

**Questions:**

- What sets your contributions apart from [The Pitfalls of Next-Token Prediction](https://arxiv.org/pdf/2403.06963)?
- Can you give me a concrete list of contributions?

## Miscellaneous

- The entire first paragraph is oddly phrased.
>Neat representations materialize when a model compresses redundancies in the data. On the
other hand, when faced with incompressible atomic facts (like the birth date of a celebrity), a
model would memorize these associations like a lookup table.

 What are ''neat representations''? The "however" also does not quite feel right here.

> We argue that this phenomenon implies a form of geometric memory in Transformers that must be contrasted with the associative memory view posited by Bietti et al. (2023).

What are geometric and associative memories?


- Figure 6 in the appendix does an excellent job at explaining the setting of this paper. It would  be great if parts of it were in the main paper.
- Figure 2: The logarithmic (?) y-axis is slightly misleading and makes 48% look like 90%.
- "However, perhaps when the target we train on **is** an in-weights path, the cheat is not easy to learn—say, due to the nature of parametric memory."
- In the list of contributions:"We argue that the emergence of the geometric memory over the associative memory does not follow directly from existing intuitions."  Arguing is  not a contribution. Do you demonstrate or prove this?
- Inconsistent subsubsections: some sections have a single subsubsection. Either use multiple ones (consistently) or do not use subsubsections.

---

> ### Author Response · Authors · 2025-11-22
> **Response**
>
> Dear review, thank you for your thoughtful review. We are glad to hear that you appreciate the empirical work in our paper (`empirical work in this paper is solid`),  and have the insight that global information is learned even if not present in the training set (`S1`).
> We have further incorporated several of your suggestions for improving the overall clarity of the writing.
>
> However, **we believe that your more major concerns, are directly answered in the main paper. Some of this confusion appears to be caused by misunderstandings of the precise terms and settings which we define in the paper.** We hope the discussion below clarifies these points; we have updated the writing significantly to ensure the definitions of these terms are more explicitly highlighted.
>
> ---
> > What are geometric and associative memories?
>
> Apologies for any ambiguity. As these are terms central to our discussion,  we had defined them at varying levels of detail at several points in the paper:
> - Both memories are described intuitively in the third paragraph and the fourth paragraph in the introduction.
> - Later, mathematically defined in `Hypotheses 3 & 4` now renamed as `Definitions 2a & 2b` for greater clarity.
> - In Sec 2.2 and 2.3 both memories are also described in more detail in prose.
>
> To make this even clearer, **we’ve now added Table 1** to describe and contrast the two notions of memory.
>
> Note that geometric memory is a new term we’ve coined, while “associative” memory is established in literature e.g., (Bietti et al., 2023; Cabannes et al., 2024a.
>
> ---
> > In the list of contributions:"We argue that the emergence of the geometric memory over the associative memory does not follow directly from existing intuitions." Arguing is not a contribution. Do you demonstrate or prove this?
>
> **This is precisely the purpose of Section 3**, where we’ve carefully demonstrated this through extensive proofs and experimental evidence! We have reworded this line. Thank you.
>
> ---
>
> >  What sets your contributions apart from The Pitfalls of Next-Token Prediction?
>
> Our paper studies a fundamentally different task from B&N’24. Although we use their graph topology, we store the graph in the model weights, while they present it in context. (For reference, please see the second paragraph and also `Sec 2.1`).
>
> Why do we use their graph topology? This topology is so hard that in-context reasoning fails in B&N’24; in contrast, it succeeds in our in-weights setting (or `in-memory` as your summary phrases it). This isolates a striking, clean instance of successful in-weights reasoning, one which has not previously been observed.
>
> ----
> > “Can you give me a list of contributions”
>
> **You will find a list of our core contributions in Section 1.1, titled “Summary of Contributions”.** If you would like a bit more detail, we can expand upon them further:
>
> 1. `Section 2:` We identify that implicit in-weights reasoning is possible in an adversarially-constructed task. We demonstrate this without step-wise supervision and other tricks used in prior literature. This presents a key contradiction to viewing deep sequence models as associative memory storage (**a contradiction never laid out in literature**).
>
> 2.
>    - (i) We spell out a clear contrast between two modes of memorization e.g., associative vs geometry. One stores local information; the other has synthesized global information not present in the training set. One is stored in an intermediate weights matrix with arbitrary embeddings; the other has carefully-constructed embeddings. **This contrast is new.**
>    - (ii)` Section 3`: Through experiments and theoretical arguments, we explain why the geometry is surprising and cannot be straightforwardly explained. This is a foundational contribution leading to a **new memorization puzzle**.  (See also common response `On the lack of a (clear) answer` for more on this puzzle.)
>
> 3. `Section 4`: We answer what this geometry is and where it comes from: a spectral bias.
>
> > Node2Vec seems a bit orthogonal
>
> This section is key to understanding the source of the geometry. We elaborate this in the common response, under `The simplicity of the Node2Vec model is a strength`.
>
> > Significance:
>
> We have now added an detailed paragraph on the “Implications” of our findings in the introduction. We hope that you and future readers find this useful.

---

> > ### Author Response · Authors · 2025-11-22
> > **Response (continued)**
> >
> > > “It is mainly a flood of hypotheses, observations and related works.“
> >
> >
> > Thank you for this valuable feedback.
> >
> > Our work ties up many fragmented lines of research in literature (e.g., papers on associative memory vs on observations about geometries vs on implicit reasoning tasks). These related works are important context. But we understand the concern and have simplified these citations a bit.
> >
> > We also understand why the “hypothesis → observation” can be confusing. Perhaps we can clarify that the “Hypothesis” are meant to be viewed as “Candidate Hypotheses” based off of intuition that is common or simple. These are presented from the most simplest/natural hypotheses first—and refuted—before making the case for the more complex ones.
> >
> > > The logarithmic (?) y-axis is slightly misleading
> >
> > Our focus is on the enormous gap between random chance accuracy and the observed accuracy of 50-100%. Without the log scale, random chance accuracy of ~0.1% would not be distinguishable from 0. To avoid confusion, we had printed the exact numbers directly in the image.
> >
> > Thanks once again for your review and for catching some of the other writing issues. We are happy to answer further questions.

---

> > > ### Comment · Reviewer_9RrG · 2025-11-24
> > >
> > > I thank the authors for their response. I have re-read the paper and it is now easier to understand. As this was my core criticism I will significantly increase my score ($2$ $\rightarrow$ $6$).
> > >
> > > Some final points:
> > > - The summary of contributions you have written in your rebuttal is more informative and direct than the one in the paper. Consider replacing the summary in the paper.
> > > - You should be more direct about claiming the geometric memory term and its contrast (rebuttal - contributions 2.I). This is the key point of the paper and I believe it could be spelled out even more directly. For example, at the beginning of the abstract you write: `We contrast this associative view against a geometric view of how memory is stored.` This sentence is mostly meaningless and does not effectively communicate your contribution.
> > > - There is a typo in line 296: `other.(see`

---

> > > > ### Author Response · Authors · 2025-11-28
> > > > **Thank you**
> > > >
> > > > Thank you for engaging with our response and re-considering our paper. We will be incorporating your constructive feedback for future versions of the work!

---

### Official Review · Reviewer_UZ7j · 2025-11-01

**Soundness:** 3
**Presentation:** 3
**Contribution:** 3
**Rating:** 4
**Confidence:** 3

**Summary:**

--------------An empirical study with insufficient analytical support-----------------------------

The paper studies how Transformers, when trained to memorize graph edges (in-weights), often learn a “geometric” parametric memory—node embeddings that reflect global relationships—rather than a purely local associative lookup structure. The authors construct an in-weights path-star task (first train on edge bigrams to put a fixed graph into the weights, then finetune on path prediction) and find that Transformers can perform nontrivial multi-hop path-finding for large graphs where in-context next-token training fails. The success contradicts standard associative-memory abstractions. To explain how local pairwise supervision can yield global geometry, the paper studies simpler 1-layer Node2Vec models and presents empirical evidence of a spectral bias: embeddings align with Fiedler-like eigenvectors of the graph Laplacian, while a time-varying coefficient matrix converges to having those eigenvectors in its null space. The authors argue a similar spectral mechanism may be responsible in Transformers, though Transformer embeddings are less “clean” than Node2Vec’s, indicating room for improvement.

Overall, the paper presents a clear phenomenon, thorough experiments, and a promising connection to spectral dynamics in simpler models.

**Strengths:**

•	Clear, analyzable phenomenon: Turning the known in-context failure for path-star graphs into an in-weights setting produces a striking contrast and isolates implicit in-weights reasoning.
•	Extensive experiments: Results span multiple graph types and sizes and include meaningful ablations (edge directions, pause tokens, first-token-only loss, etc.). Visualizations and heatmaps strengthen the empirical case.
•	Theoretical connection: Mapping the emergence of geometry to spectral bias in Node2Vec-style learning is a useful and plausible explanatory route.
•	Useful framing: Presenting associative vs. geometric memory as competing data structures is intuitive and helpful for framing future research.
•	Reproducibility effort: The paper includes many implementation details, hyperparameters, and procedural choices; the appendices are substantial.

**Weaknesses:**

1.	Theoretical rigor is limited.
o	The claim that Node2Vec dynamics converge to embeddings aligned with Fiedler vectors and that the coefficient matrix has those vectors in its null space is primarily empirical. The paper acknowledges the lack of a formal proof; still, the scope and assumptions of the empirical claim should be clarified (e.g., dependence on initialization, optimizer settings, sampling, etc.).
2.	Generality and applicability to natural language are unclear.
o	The experiments are on symbolic graph tasks and specific topologies. It is not yet clear whether the geometric memory phenomenon plays a comparable role in real-world language-model memorization or knowledge integration.
3.	Explanation for why Transformers avoid associative solutions is incomplete.
o	The paper effectively shows simple statistical/architectural/supervisory explanations are insufficient, but lacks a direct diagnostic showing why gradient descent prefers geometric solutions in practice. More direct analysis of optimization trajectories or mechanisms would strengthen this point.
4.	Some training choices require deeper analysis.
o	Pause tokens, mixed forward/backward edge supervision, and the particular interleaving strategy are important in practice; their exact roles (and whether they generalize) need more principled investigation.
5.	The link between Node2Vec and Transformer remains somewhat speculative.
o	Transformers include attention, deep layers, residual connections and layer-norm; how these aspects alter or obstruct the spectral dynamics observed in Node2Vec isn’t fully explored. Additional ablations to bridge the gap would be helpful

**Questions:**

Experiments & results

•	Stability: Provide multi-seed runs and report variance (e.g., success rate, epochs-to-convergence). Many plots show single runs.

•	Learning dynamics: The result that token accuracies rise “in tandem” is informative. Please show whether this pattern is robust to changes in learning rate, batch size, optimizer, or initialization.

•	Pause tokens: Quantify the relationship between number of pause tokens and required reasoning depth. Does increasing model depth (more layers) substitute for pause tokens?

•	Mixed edge supervision: Present a fine-grained ablation where the fraction of backward edges is varied continuously (e.g., 0%→100%) to show how performance scales.

•	Quantitative geometry metrics: In addition to heatmaps and UMAPs, report objective metrics (e.g., intra-path vs. inter-path cosine separation, Silhouette score, alignment with true Fiedler vectors) with error bars so comparisons (Transformer vs Node2Vec vs associative) are objective.

Theory & analysis

•	Node2Vec dynamics: Move the full derivation of Lemma 2 into the appendix (if not already) and explicitly list all assumptions (batching, softmax normalizations, self-probabilities, sampling, optimizer choices). Clarify when the empirical “self-stabilizing” dynamics are expected to hold and when they may fail.

•	Proposition 1: The bits and l2 norm arguments are useful. Flesh out boundary conditions—e.g., effect of weight tying, multiple outputs per input, or non-uniform degree distributions.

•	Transformer mechanism: Consider intermediate ablations that simplify the Transformer toward Node2Vec—e.g., single-layer Transformer with only embedding+unembedding, linear attention, or frozen attention weights—to trace whether spectral alignment persists and how components contribute.

---

> ### Author Response · Authors · 2025-11-21
> **Response**
>
> Thank you for your extensive feedback. We appreciate the care with which you laid out your thoughts. However, **we strongly disagree with several of the stated weaknesses**, which we detail below. Others limitations, we acknowledge, such as the lack of a formal theory, as we note explicitly in `Section A`.
>
> > “Explanation for why Transformers avoid associative solutions is incomplete.”
>
> Our contribution lies in formulating a new foundational question; we've also put forth possible answers and refuted them. We've expanded on why this is not a weakness in the common response under `On the lack of a (clear) answer`.
>
> > “The link between Node2Vec and Transformer remains somewhat speculative.”
>
> On the contrary, the Node2Vec analysis neatly isolates the core of the phenomenon in a Transformer. We clarify this in the common response with updated experiments. Please see `The simplicity of the Node2Vec model is a strength`.
>
> > “variance across seeds”
>
> Across multiple random seeds, our models achieve accuracy that is several orders of magnitude higher than random chance. Thus our general findings are unaffected by any variance.
>
> > Experiments on pause tokens and reverse edge
>
> Thanks for the suggestion! We consider these details to be less core to the main message of the paper about the competition between two types of parametric memories, but we agree there is room for further investigation. We have added some experiments on reverse edge supervision in Sec. F. We hope the pre-existing sensitivity analysis of pause tokens in Sec. F  suffices for the main message present paper. This indeed would benefit from separate investigations in the future.
>
> > Proposition 1: boundary conditions:
>
> We have updated this, thank you.
>
> Please let us know if you've any further questions.

---

> > ### Comment · Reviewer_UZ7j · 2025-11-27
> > **Could you explain the conditional observations and their underlying connections?**
> >
> > The contributions are not formally presented.
> >
> > Could you explain the conditional observations and their underlying connections?
> >
> > Could you provide more formal theoretical conclusions?
> >
> > For example, could you provide an approximate parameterized relationship between complexity and scale?

---

### Author Response · Authors · 2025-11-21
**Response to common concerns**

We thank the reviewers for their feedback.  We have updated the paper with new experiments and improved presentation. Below, we clarify the common criticisms raised.

---

### **On the lack of a (clear) answer**

Multiple reviewers are concerned that we do not give a clear answer to the problem we have posed (`UZ7j, eCS4`). We understand that a lack of an answer can be dissatisfying. However:

**Lots of good research is about asking the right questions, which can help steer the community into new lines of inquiry—this is the main purpose of our work.**

- Our paper highlights a new contrast between modes of memorization.
- It formulates a foundational **memorization puzzle**: _why does the gradient-descent-trained model memorize via a geometry instead of a lookup table, even when the geometry is not inherently more succinct in bit complexity, and even with just local supervision?_
- We've not just asked a question, but also gone over several natural explanations to this phenomenon, and provide careful experiments/theoretical justifications to rule them all out (`Section 3`).

Such a contribution can be widely influential, such as in papers like “understanding deep learning requires rethinking generalization” [Zhang et a., 2017 https://arxiv.org/abs/1611.03530). They posed  “the generalization puzzle”, but they did not solve it, and this challenge was taken up by 1000s of papers. Analagously, we pose a “memorization puzzle in sequence modeling” which we hope will inspire many new lines of theoretical inquiry, and also practical work improving Transformer geometry.

-----

### **The simplicity of the Node2Vec model is a strength**

We understand that the lack of an attention layer in our Node2Vec analysis appears too simplistic on the face of it (`UZ7j 9RrG, 2EuG`). Let us clarify why, on the contrary, this simplicity is a strength: it cleanly isolates the source of the geometries we see.

- First, we have added **new experiments** which show similar geometries in Mamba and neural networks. Thus attention layers are not critical to the geometry. (`E.1 and Fig 12-15`)
- The node2vec model is simply a Transformer model with the same loss, same dataset, but without the intermediate attention/MLP layers (as noted in `Sec 4 para 1`). Thus, our analysis implies that **there is a spectral bias which is fundamental to any gradient-descent-trained deep sequence model on local co-occurrences.**
- Third, an analysis for even such simple node2vec style models on cross-entropy loss has been a highly non-trivial long-standing open question. So far this has only been studied in settings even simpler than ours, such as on a simpler loss, or with explicit rank constraints (See `Related Work Sec 5`).

----

### **The  “Hypothesis”, “Observation” style of presentation**

Reviewer `9RrG` understandably finds our presentation style confusing since we put forth hypotheses only to rule them out later on. We must clarify that our “Hypotheses” must be viewed as as “Candidate Hypotheses”, based off of existing intuitions, that may or may not be true. Our intention was to ensure simple explanations are ruled out first, which is the nature of our result here.

---

### Note · Authors · 2025-12-12

**Comment:**

After discussions between the authors, we believe it is best to withdraw the submission at this point. We thank the reviewers for their feedback, and `9RrG` in particular for their second read of the paper; we also thank the AC(s) for their time and effort in managing this year's challenging reviewing process.

**Withdrawal Confirmation:**

I have read and agree with the venue's withdrawal policy on behalf of myself and my co-authors.